# MANGO: Natural Multi-speaker 3D Talking Head Generation via 2D-Lifted Enhancement

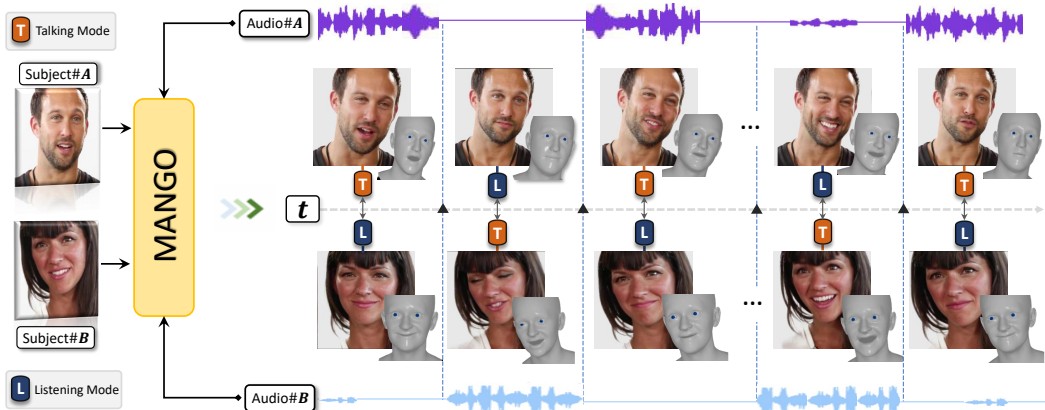

Figure 1: The illustration shows A and B conversing, the blue box with 'L' indicates that this person is listening, while the orange box with 'T' indicates that this person is talking. The 3D mesh sequences as well as 2D video of A can be synthesized from their conversational audio and a reference image of A, with the same process applying to B.

## ABSTRACT

Current audio-driven 3D head generation methods mainly focus on single-speaker scenarios, lacking natural, bidirectional listen-and-speak interaction. Achieving seamless conversational behavior, where speaking and listening states transition fluidly remains a key challenge. Existing 3D conversational avatar approaches rely on error-prone pseudo-3D labels that fail to capture fine-grained facial dynamics. To address these limitations, we introduce a novel two-stage framework ***MANGO***, which leveraging pure image-level supervision by alternately training to mitigate the noise introduced by pseudo-3D labels, thereby achieving better alignment with real-world conversational behaviors. Specifically, in the first stage, a diffusion-based transformer with a dual-audio interaction module models natural 3D motion from multi-speaker audio. In the second stage, we use a fast 3D Gaussian Renderer to generate high-fidelity images and provide 2D-level photometric supervision for the 3D motions through alternate training. Additionally, we introduce MANGO-Dialog, a high-quality dataset with over 50 hours of aligned 2D-3D conversational data across 500+ identities. Extensive experiments demonstrate that our method achieves exceptional accuracy and realism in modeling two-person 3D dialogue motion, significantly advancing the fidelity and controllability of audio-driven talking heads.

## 1 INTRODUCTION

In recent years, considerable research has been dedicated to 3D talking head generation, with a particular emphasis on audio-driven 3D motion synthesis for applications in virtual reality, film production, education, *etc.* However, current talking head models are typically constrained to either a *speaking* or *listening* mode, lacking the ability to transition smoothly and realistically between

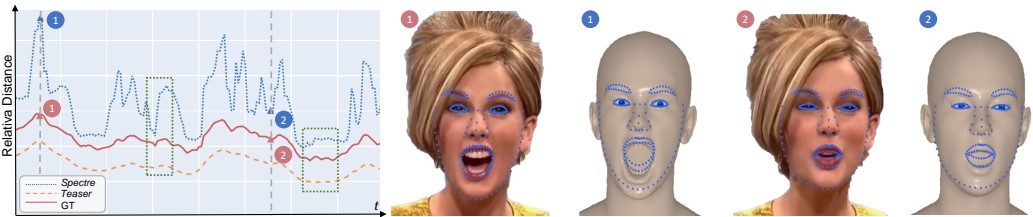

Figure 2: Limitations of existing 3D face reconstruction for training conversational talking head. The estimated 3D data either exhibits over-smoothed mouth movements (orange dashed curve) compared to actual lip movements (red curve), or shows exaggerated and noisy movements (blue dotted curve). Such visual misalignments are illustrated on the right. The red curve is derived by calculating the distance between the corresponding key points of the manually annotated upper and lower lips, which effectively describes the actual mouth movement. The blue curve represents the average distance between the corresponding key points of the upper and lower lips, calculated after projecting the 3D mesh reconstructed by Spectre back into 2D space. The orange curve shows the results from Teaser.

these two states. Specifically, *speaker-only* models Zhang et al. (2023); Liu et al. (2024); Xing et al. (2023); Fan et al. (2022) excel at generating lip movements that are highly synchronized with speech but fall short in producing natural listening feedback. Conversely, *listener-only* models Ng et al. (2022); Song et al. (2023) can generate convincing attentive responses but are incapable of speaking. Yet in real-world human-computer interaction, both speech generation and responsive listening are equally critical for digital humans. This highlights the importance of advancing conversational head generation, a unified framework capable of engaging in fluid interaction.

Recently, INFP Zhu et al. (2025) was the first to propose a multi-speaker 2D talking head generation framework. This approach utilizes an end-to-end audio-to-video generation, which cannot achieve fine-grained control over individual components such as mouth motion or identity consistency. Moreover, it cannot be applied to 3D talking head generation. DualTalk Peng et al. (2025), on the other hand, proposes a unified framework for dual-speaker interaction 3D talking head generation, using estimated 3D meshes as ground-truth to constrain the final output. However, in conversational scenarios, lip movements become more complex to model due to the dynamics of interaction Peng et al. (2025); Zhu et al. (2025). We observed that recent 3D face reconstruction methods tend to exhibit significant misalignment and inaccuracy between the meshes, particularly in the lip region, as shown in Fig. 2. This misalignment fails to provide adequate supervision for precise 3D talking head generation. In contrast, 2D visual data contains clear and direct interaction patterns in dialogues. Leveraging them as supervision helps compensate for the inaccuracies introduced by 3D motion tracking.

Motivated by the above observations, we propose a two-stage network shown in Fig. 3 that leverages image-level supervision to generate multi-speaker talking heads, aiming to improve the accuracy of 3D motion generation. In the first stage, our diffusion-based motion generation network generates a sequence of 3D motion parameters from the input audio. To achieve both fidelity and training efficiency, we introduce a Gaussian splatting-based renderer in the second stage, which synthesizes video frames by rendering the predicted meshes with a given reference image. These two stages are first pretrained separately and then combined for joint training. This design fully utilizes image-level supervision to compensate for inaccuracies in 3D pseudo-labels by tracking, thereby further enhancing the precision of 3D motion generation. In addition, we introduce a temporally-aligned and 2D-3D aligned multi-speaker dialogue dataset MANGO-Dialog. It contains high-quality videos covering various scenarios, such as daily communication, deep emotional interaction, spoken language teaching, and live interviews. After we acquired 2D conversational videos, we performed tracking and fitting on the dataset, obtaining pseudo 3D motion labels and camera parameters, respectively. In summary, our contributions are as follows:

**(1)** We propose a novel two-stage conversational generation framework that leverages 2D images as supervision to compensate for inaccuracies in 3D tracking results, thereby providing more reliable guidance for generating multi-speaker 3D talking heads.

**(2)** In the first stage, we design a diffusion-based multi-audio fusion module to model the motion distribution; in the second stage, we employ a 3D Gaussian Renderer that utilizes both a reference

image and the motion sequence from the first stage to generate the final output video. The two stages are first pretrained and then combined for joint training.

**(3)** We present a high-quality, photometrically-aligned and well-synced conversion dataset. Our experimental results demonstrate that our method outperforms existing approaches in multi-speaker 3D conversation tasks, particularly in terms of mouth movement accuracy and visual alignment with 2D target videos.

## 2 RELATED WORK

### 2.1 3D TALKING HEAD GENERATION

The field of speech-driven 3D talking head generation Sun et al. (2024); Yu et al. (2024) continues to garner significant research attention. Existing methodologies in this domain typically follow two distinct approaches. The first approach Fan et al. (2022); Xing et al. (2023); Sun et al. (2024) utilizes acoustic features, such as MFCCs or representations derived from pretrained speech models Baevski et al. (2020); Hannun et al. (2014); Hsu et al. (2021), mapping them to either 3D Morphable Model (3DMM) parameters Cao et al. (2013); Paysan et al. (2009); Li et al. (2017) or a 3D mesh Cudeiro et al. (2019); Haque & Yumak (2023); Richard et al. (2021b) to achieve decoupled expression and motion control. However, a major limitation lies in the lack of true 3D ground-truth labels, forcing methods to rely on 3D pseudo-labels generated by 3D reconstruction techniques Wang et al. (2024); Feng et al. (2021); Retsinas et al. (2024); Liu et al. (2025), whose accuracy is consequently constrained by the absence of genuine 3D supervision. Alternatively, the second category of methods Yu et al. (2024); Qian et al. (2024); Wang et al. (2025) employs a pure 3D Gaussian pipeline, where offsets of Gaussian attributes are predicted through spatial-audio interaction to render the desired deformation effects. Nevertheless, a critical drawback of these techniques is their current restriction to a single person per model, hindering their generalization ability to arbitrary identities. Additionally, due to the highly disentangled and controllable nature of 3D parameters, many 2D facial animation methods also utilize 3D motion parameters to model facial movements. Ren et al. (2021); Yin et al. (2022); Zhang et al. (2023) learn the mapping from audio to 3D parameters and use these parameters as control signals to generate talking heads, thereby achieving highly controllable results.

### 2.2 MULTI-SPEAKER AUDIO-DRIVEN MOTION GENERATION

With the rapid development of audio-driven digital human generation, audio-driven motion generation for multi-speaker interaction has emerged as an important research direction. Most existing methods support only a single function (speaking or listening), failing to achieve natural transitions between these two states. Zhu et al. (2025) pioneered a speech-driven 2D talking head generation method for multi-speaker interaction by mapping speech to visual latent codes containing conversational semantics to animate a static image. However, its latent space does not offer explicit control, and any fine-tuning requires re-inference from the audio input. Yu et al. (2025) employed Stable Diffusion and Appearance Reference Net, using speech as condition along with LLM-generated labels to produce three states (listening, speaking, and communicating). However, this approach struggles to capture inter-speech relationships, resulting in insufficient representation of interactive semantics in generated videos. Kong et al. (2025) implemented multi-stream audio injection through Label Rotary Position Embedding (L-RoPE), computing cross-attention between video latent space and speech embeddings using DIT architecture. While achieving realistic results, it demands excessive computational resources for training and inference. In the field of 3D interactive generation, DualTalk Peng et al. (2025) first realized speech-driven 3D digital human generation for multi-speaker interaction with natural state transitions, but its mouth movement modeling lacks precision due to reliance solely on 3D pseudo-label supervision. Qi et al. (2025) achieved 3D human motion modeling via a temporal interactive module, Ghosh et al. (2025) extracted motion token sequences from speech through Hierarchical Masked Modeling, and Mughal et al. (2024) performed multi-speaker interaction synthesis using diffusion models conditioned on multimodal inputs like text and speech. None of these methods addressed the alignment between 3D and 2D generation. Our work is the first to incorporate 2D supervision in 3D conversational head generation, significantly improving the accuracy of 3D motion generation.

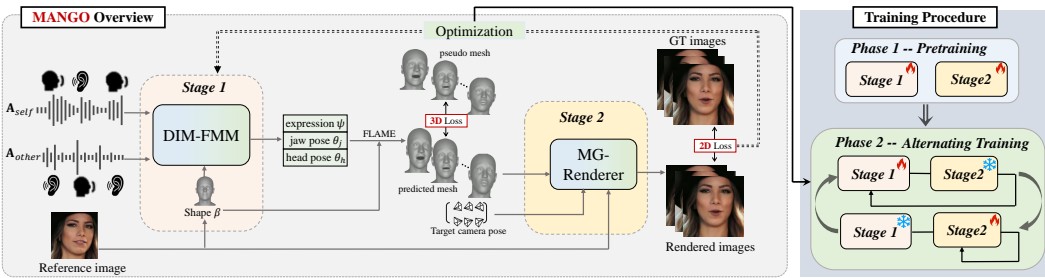

Figure 3: The overall pipeline of MANGO. Our method first generates 3D facial motions from speech via **DIM-FMM** (stage1: audio-to-mesh), and then synthesizes images with **MG-Renderer** (stage2: mesh-to-image), where 2D supervision further refines the 3D motion.

## 3 METHOD

### 3.1 OVERVIEW

As illustrated in Fig. 3, the proposed MANGO consists of two stages. Specifically, given an agent audio $\mathbf{A}_{self}$, a equal-length conversation partner audio $\mathbf{A}_{other}$, the shape parameter $\beta$ of the agent speaker, and an agent speaker indicator $\mathbb{I}_{self}$ indicating whether the current speaker is speaking. The proposed first-stage diffusion model generates the corresponding motion parameter sequences $\mathbf{X} = \{\psi, \theta_j, \theta_h\}$ under the above conditions. Here $\{\psi, \theta_j, \theta_h\}$ indicate the expression, jaw pose, and head rotation parameters, respectively. Then we can generate the facial 3D geometry through a 3D morphable model Li et al. (2017), *i.e.*,

$$\mathbf{V} = \text{FLAME}(\beta, \psi, \theta_j, \theta_h), \tag{1}$$

where $\mathbf{V} \in \mathbb{R}^{w \times 5023 \times 3}$, $w$ is the length of generated frames. Subsequently, based on a reference image $I$, 3D facial geometries $\mathbf{V}$, and a pre-defined camera pose $\mathbf{R}$, we adopt a 3D Gaussian renderer to generate the final 2D video.

### 3.2 MOTION GENERATION WITH CONVERSATION AUDIO

**Dual-audio Interaction module (DIM).** Given an agent audio $\mathbf{A}_{self}$, and the audio of other speaker $\mathbf{A}_{other}$, we propose a dual-audio infusion module (**DIM**) as shown in the left of Fig. 4 to learn the joint audio representation. We extract the audio features $\mathbf{H}_{self}$ and $\mathbf{H}_{other}$ using a pre-trained audio encoder $\mathcal{E}$ Hsu et al. (2021):

$$\mathbf{H}_{self} = \mathcal{E}(\mathbf{A}_{self}), \quad \mathbf{H}_{other} = \mathcal{E}(\mathbf{A}_{other}), \tag{2}$$

where $\mathbf{H}_{self}$ and $\mathbf{H}_{other}$ are the audio features of the agent speaker and the other speaker, respectively. The audio features $\mathbf{H}_{self}$ and $\mathbf{H}_{other}$ are then fed to a Transformer encoder to capture long-range dependencies and intricate interaction patterns between those two speakers. However, incorporating the other speaker's audio may dilute the contribution of the speaking agent's audio cues. To preserve motion-speech synchronization for the agent speaker, we adopt a residual connection by adding $\mathbf{H}_{self}$ back to the fused representation. The final dual audio feature $\mathbf{H}_{dual}$ is generated by:

$$\mathbf{H}_{dual} = \text{Transformer}(\mathbf{H}_{self}, \mathbf{H}_{other}) \oplus \mathbf{H}_{self}. \tag{3}$$

The fused audio feature are then concatenated with an agent speaker indicator $\mathbb{I}_{self}$, which is a binary vector indicating whether the agent speaker is speaking. The concatenated features are then fed into a linear layer to obtain the fused audio feature representation $\mathbf{H}_{fuse}$.

**Fused-audio Motion Generation Model (FMM).** Based on $\mathbf{H}_{fuse}$, we design a diffusion-based model to predict the synchronized motion sequences. The fused-audio feature $\mathbf{H}_{fuse}$ is fed into a diffusion transformer to generate the corresponding motion parameter sequences $\mathbf{X} = \{\mathbf{x}_1, \mathbf{x}_2, ..., \mathbf{x}_T\}$, where $\mathbf{x}_i \in \mathbb{R}^{56}$. In the forward diffusion process, Gaussian noise $\mathbf{z}$ is added to the initial data sample $\mathbf{X}^0 = \{\mathbf{x}_0^0, \mathbf{x}_1^0, ..., \mathbf{x}_T^0\}$ according to a variance schedule. Eventually, the data distribution converges to a standard normal distribution, which can be represented as $q(\mathbf{X}^N | \mathbf{X}^0)$. In the reverse process, we use the distribution $q(\mathbf{X}^{n-1} | \mathbf{X}^n)$ to gradually recover the original sample from noise. To predict this distribution $q(\mathbf{X}^{n-1} | \mathbf{X}^n)$, we employ a denoising network to directly predict the

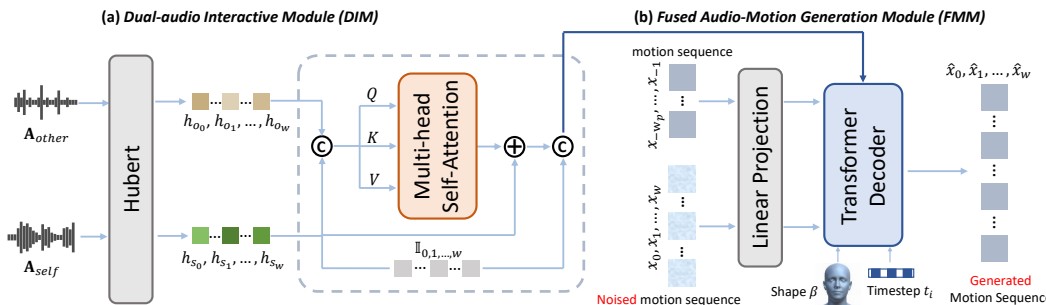

Figure 4: Dual-audio interactive module (**DIM**) and fused audio motion generation module (**FMM**). The speech signals $\mathbf{A}_{self}$ and $\mathbf{A}_{other}$ are fed into Hubert for semantic features $\mathbf{h}^s_{0:w}$ and $\mathbf{h}^o_{0:w}$. These features are concatenated and fed into a multi-head self-attention module, which is then combined with $\mathbf{h}^s_{0:w}$ via residual connection, followed by concatenation with the agent speaker indicator $\mathbb{I}$. These features are then fed into **FMM** for motion $\hat{\mathbf{X}}_{0:w}$ generation.

clean sample from the noisy sample, rather than stepwise predicting the noise at each step. This approach enables the introduction of geometric loss, providing more precise constraints for facial motion and significantly benefiting motion generation Sun et al. (2024); Tevet et al. (2022). Specifically, our **FMM**, as shown in the right part Fig. 4, performs the denoising process as follows: for the predicted fused-audio feature $\mathbf{H}_{fuse}$, we divide it into frame blocks of $w$ frames, and use $\mathbf{H}_{0:w}$ as the input for each block. For each window, at diffusion step $n$, the denoising network takes as input the previous and current fused audio features $\mathbf{H}_{-w_p:w}$, the real previous motion sequence $\mathbf{X}^0_{-w_p:0}$, and the current noisy sequence $\mathbf{X}^n_{0:w}$ sampled from $q(\mathbf{X}^n_{0:w}|\mathbf{X}^0_{0:w})$. In addition to the fused-audio feature, we incorporate the FLAME shape parameter $\beta$ shared across all windows. Therefore, the final input and output format of the denoising network can be represented as:

$$\hat{\mathbf{X}}^0_{-w_p:w} = \mathrm{DiT}(\mathbf{H}_{-w_p:w}, \mathbf{X}^0_{-w_p:0}, \mathbf{X}^n_{0:w}, n, \beta). \tag{4}$$

In terms of loss optimization, the loss for the first stage consists of three parts:
1) **Parameter Loss.** We calculate the L2 loss between the denoised motion and the ground-truth motion at each step, directly constraining the the predicted parameter distribution to be close to the real distribution:

$$\mathcal{L}_{param} = ||\hat{\mathbf{X}}^0_{-w_p:w} - \mathbf{X}^0_{-w_p:w}||^2_2, \tag{5}$$

since jaw pose parameters $\theta_j$ are especiallt important for lip movements, we also addtionally add a loss on jaw pose parameters:

$$\mathcal{L}_{jaw} = ||\hat{\theta}_{j,-w_p:w} - \theta_{j,-w_p:w}||^2_2, \tag{6}$$

2) **3D Loss.** we convert the parameters into zero-head-posed 3D mesh sequences, then we get $\mathbf{V}_{-w_p:w_p} = \mathrm{FLAME}(\beta, \mathbf{X}^0_{-w_p:w_p})$ and $\hat{\mathbf{V}}_{-w_p:w_p} = \mathrm{FLAME}(\beta, \hat{\mathbf{X}}^0_{-w_p:w_p})$ and compute geometric loss directly in the 3D space.
3) **Stability Loss.** We use velocity loss to improve temporal consistency, and smooth loss to regularize excessive motion amplitudes and prevent abrupt changes. More details will be presented in the supplementary material. The overall loss for the first stage is:

$$\mathcal{L}_{\mathrm{stage1}} = \mathcal{L}_{\mathrm{param}} + \lambda_{\mathrm{jaw}}\mathcal{L}_{\mathrm{jaw}} + \lambda_{\mathrm{vert}}\mathcal{L}_{\mathrm{vert}} + \lambda_{\mathrm{vel}}\mathcal{L}_{\mathrm{vel}} + \lambda_{\mathrm{smooth}}\mathcal{L}_{\mathrm{smooth}}. \tag{7}$$

### 3.3 Image Synthesis with Meta Gaussian Renderer

To enhance the synchronization and fidelity of the generated results, we propose a 3D meta Gaussian renderer (**MG-Renderer**) to render the above generated motion sequences into high-fidelity 2D images, which was then supervised using ground-truth images. Specifically, given $\hat{\mathbf{X}}^0_{0:w}$, we randomly sample $n$ frames from this window sequence to get $S = \{\hat{\mathbf{x}}_0, \hat{\mathbf{x}}_1, ..., \hat{\mathbf{x}}_n\}$. Through FLAME, we obtain the vertex sequence $V = \{\hat{\mathbf{v}}_0, \hat{\mathbf{v}}_1, ... \hat{\mathbf{v}}_n\}$. We randomly sample an image of the agent speaker as a reference image $I_r$, which is fed into Dinov2 Oquab et al. (2023) for the appearance feature. Besides, we use target camera parameters $R$ which is a $4 \times 4$ matrix including translation

and rotation scale, to project the vertices into 2D image plane. For each $\hat{\mathbf{x}}_i$, the process through our carefully designed Meta Gaussian Renderer $\mathcal{R}$ can be described as:

$$\hat{\mathbf{I}}_i = \mathcal{R}(\hat{\mathbf{v}}_i, I_r, R). \tag{8}$$

### 3.3.1 META GAUSSIAN RENDERER

In our Gaussian Renderer $\mathcal{R}$, we model faces in a canonical space using 3D Gaussians Kerbl et al. (2023). Each Gaussian is defined by its position, rotation, scale, opacity, and a latent appearance vector: $G = \{\mu, r, s, \alpha, c\}$. The meta Guassians are constructed from two components: **(1)** Template Gaussians derived from the FLAME model $G_T$ which are relatively sparse and responsible for representing the overall texture and geometry; and **(2)** UV Gaussians attached to the triangulated mesh $G_{UV}$, which are used to encode fine-grained details. The detailed construction of each Gaussian and Renderer $\mathcal{R}$ are provided in the supplementary material A.3.

### 3.3.2 IMAGE-LEVEL SUPERVISION

After the above processing, we obtain all the Gaussians of the reference image $I_r$. Then we replace the positions of the Gaussians corresponding to $I_r$ with those of $\hat{\mathbf{v}}_i$ to generate the animated Gaussians, while keeping all other attributes unchanged. During rendering, we splat the animated Gaussians to produce a coarse feature map $F_{raw}$, where the first three channels correspond to a coarse RGB image $\hat{I}_{raw}$. To enhance texture details, we feed this feature map $F_{raw}$ into a StyleUNet-based refiner, ultimately producing the final image $\hat{\mathbf{I}}_i$ with enhanced features such as detailed teeth textures.

**Loss Function**. For the sequence $\hat{\mathbf{X}}^0_{0:w}$ generated in the first stage, we randomly sample $S$ and obtain $\hat{\mathbf{I}}_{0:n} = \{\hat{\mathbf{I}}_0, \hat{\mathbf{I}}_1, ..., \hat{\mathbf{I}}_n\}$ through the rendering process described above. Similar to previous methods, we compute image-level losses for each frame in $\mathcal{R}$, mainly including photometric loss $\mathcal{L}_{pho}$ and VGG perceptual loss $\mathcal{L}_{per}$, to ensure consistency between the rendered images and the ground truth.

$$\mathcal{L}_{stage2} = \lambda_{pho}\mathcal{L}_{pho} + \lambda_{per}\mathcal{L}_{per}. \tag{9}$$

**Two-phase Training Strategy.** Our total training process contains two phases:
1) Training phase 1. To ensure that we can learn fully decoupled FLAME parameters from the audio, we first train the first stage with $\mathcal{L}_{stage1}$. Additionally, we also train the second stage separately with $\mathcal{L}_{stage2}$ to prevent the potential catastrophic impact that from-scratch training have on the first stage.
2) Training phase 2. In order to introduce the image-level loss and reduce the effect of the guassian renderer compensating between these two stages Liu et al. (2025), we alternate the training between the first and second stages. At this point, the loss function for the first stage is:

$$\mathcal{L}_J = \mathcal{L}_{stage1} + \mathcal{L}_{stage2}, \tag{10}$$

where $\mathcal{L}_{stage2}$ provides a more accurate optimization path for the first stage. After one iteration, we still apply $\mathcal{L}_{stage2}$ to optimize the second stage for better converging.

### 3.4 MANGO-DIALOG DATASET

To support the training of our multi-speaker framework, we constructed a large-scale dataset of dual-speaker dialogues (MANGO-Dialog). For diversity and authenticity, we collected dialogue videos over various daily conversation scenarios, such as emotional communication, casual dialogue, interview connections, etc. MANGO-Dialog contains more than 5,000 dialogue clips ranging from 30 seconds to 2 minutes, with a total duration of 50 hours and covering 500 speakers. The clip length setting of 30 seconds to 2 minutes ensures the inclusion of listening-speaking state transitions, which proves to be beneficial for our task. Each dialogue clip ensures that both people appear on screen simultaneously, allowing us to obtain any facial changes of both participants.

Our conversational video processing pipeline includes three steps: (1) using TackNet Tao et al. (2021) to separate speech segments and assign them to the corresponding speaker; (2) applying Spectre Filntisis et al. (2022) for 3D FLAME motion parameter tracking; and (3) refining camera parameters via keypoint-based alignment optimization.

Finally, all clips are divided into three parts: training, validation, and test sets, containing 4,300 clips, 400 clips, and 200 clips, respectively. To evaluate the model's generalization ability, the identities in the testset were unseen during training. We used the testset for subsequent metric evaluation.

Table 1: Quantitative comparisons of the comparative methods on mesh accuracy in our MANGO-Dialog testset and Dualtalk testset.

| Version | MANGO-Dialog Testset | | | | | DualTalk Testset | | | | |
|---|---|---|---|---|---|---|---|---|---|---|
| | LVE↓ | MVE↓ | MOD↓ | MTM↓ | SLCC↑ | LVE↓ | MVE↑ | MOD↓ | MTM↓ | SLCC↑ |
| FaceFormer | 3.276 | 1.754 | 1.463 | 4.812 | 0.623 | 3.186 | 1.685 | 1.442 | 4.769 | 0.632 |
| CodeTalker | 3.445 | 1.638 | 1.477 | 4.793 | 0.638 | 3.258 | 1.612 | 1.456 | 4.703 | 0.641 |
| DiffPoseTalk | 2.694 | 1.542 | 1.391 | 4.781 | 0.645 | 2.574 | 1.503 | 1.347 | 4.692 | 0.649 |
| ARTalk | 2.452 | 1.521 | 1.304 | 4.532 | 0.662 | 2.368 | 1.425 | 1.289 | 4.487 | 0.671 |
| DualTalk | 2.083 | 1.382 | 1.228 | 4.321 | 0.707 | 2.021 | 1.226 | **1.145** | 4.297 | 0.721 |
| Ours | **1.741** | **1.225** | **1.096** | **4.015** | **0.791** | **1.894** | **1.182** | 1.162 | **4.024** | **0.764** |

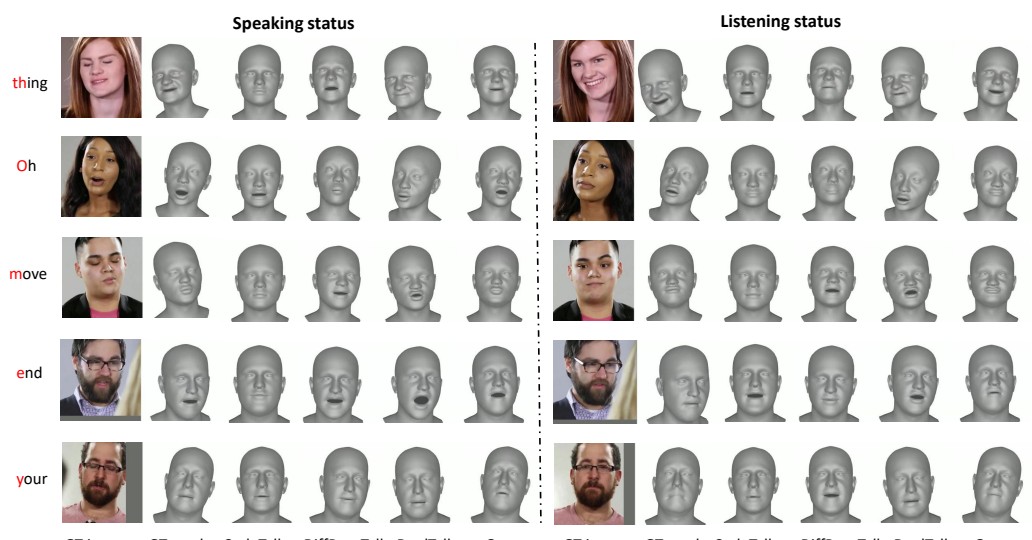

Figure 5: Visual comparison of the 3D conversational talking head generation results with SOTA methods on our MANGO-Dialog testset.

## 4 EXPERIMENT

### 4.1 EXPERIMENTAL SETUP

**Evaluation Metrics.** In our experiments, we use **LVE** Richard et al. (2021a), **MVE**, **MOD** Xing et al. (2023), **MTM** Chae-Yeon et al. (2025) and **SLCC** Chae-Yeon et al. (2025) to evaluate our 3D mesh modeling and use **PSNR**, **SSIM** , **LPIPS** Zhang et al. (2018), **LSE-D**, **LSE-C** Liang et al. (2024) to evaluate our generated 2D image quality. To comprehensively evaluate the interaction capability, we utilize **FD** to measure the realism of the listener and speaker states separately, and use **SID** to evaluate the diversity of facial expressions and head movements. The details of all metrics and our implementation details are provided in the supplementary material A.5 and A.6.

**SOTA Methods.** We compare our method with several SOTA methods: single-speaker 3D facial animation methods FaceFormer Fan et al. (2022), CodeTalker Xing et al. (2023), DiffPoseTalk Sun et al. (2024), ARTalk Chu et al. (2025) , and the recent dual-speaker 3D facial animation method DualTalk Peng et al. (2025). We also compare with single-speaker 2D talking head generation methods SadTalker Zhang et al. (2023) and AniTalker Liu et al. (2024), to evaluate our 2D generation capability. For the single-speaker methods, we mutes the speech of the other conversational partner.

### 4.2 QUANTITATIVE AND QUALITATIVE RESULTS

#### 4.2.1 QUANTITATIVE RESULTS

We evaluated our 3D performance on our MANGO-Dioglog testset and the only available 3D conversion dataset, DualTalk. As shown in Tab. 1, our method outperforms all baselines across all metrics in these two datasets, especially in LVE and MVE, indicating more accurate modeling of

Table 2: MANGO outperforms all baselines across most metrics, indicating superior realism and diversity in generated animations. 'L' stands for Listener segments, and 'S' stands for Speaker segments.

| Method | FD ↓ | | | | | | SID ↑ | | |
|---|---|---|---|---|---|---|---|---|---|
| | S-FD (exp) | S-FD (jaw) | S-FD (pose) | L-FD (exp) | L-FD (jaw) | L-FD (pose) | SID-pose ↑ | SID-exp ↑ | SID-jaw ↑ |
| DiffPoseTalk | 23.86 | 2.89 | 3.61 | 18.39 | 3.56 | 4.79 | 1.47 | 1.96 | 1.73 |
| DualTalk | **21.91** | 3.06 | 3.83 | 12.94 | 2.12 | 3.23 | 1.48 | 2.17 | 1.98 |
| Mango(Ours) | 22.37 | **2.75** | **3.54** | **11.93** | **1.99** | **2.78** | **1.53** | **2.23** | **2.26** |

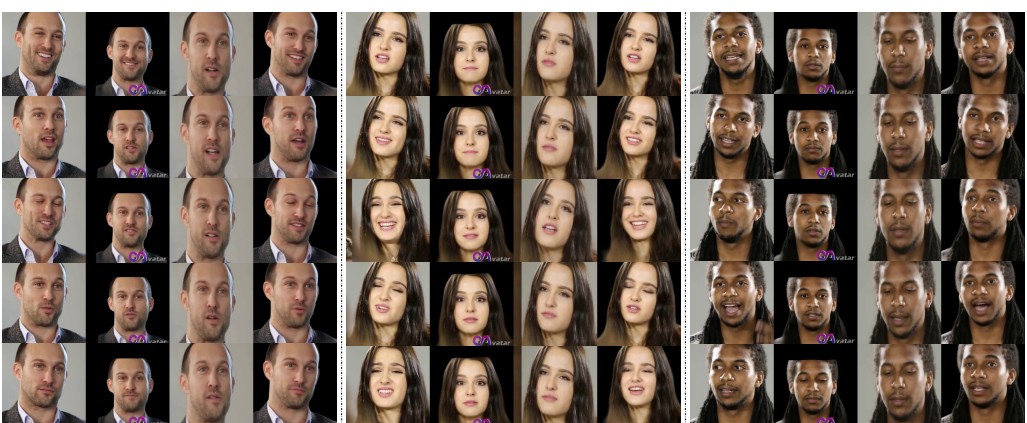

GT image  ARTalk  SadTalker  Ours          GT image  ARTalk  SadTalker  Ours          GT image  ARTalk  SadTalker  Ours

Figure 6: Visual comparison of the 2D conversational talking head generation results with SOTA methods on MANGO-Diolog testset.

frame-wise facial vertices, particularly mouth vertices. This advantage mainly stems from our dual-audio fusion module, which effectively captures the mapping between speech semantics and mouth movements, as well as the joint training of the two stages, which provides stronger supervision for motion generation. The lower MOD and MTM further confirms the improved lip-sync accuracy of our approach, while a higher SLCC indicates stronger correlation between the mesh generated by our method and vocal intensity.

Due to the lack of publicly available 2D conversation datasets, we conducted the 2D evaluation solely on the testset of MANGO-Dialog, and the results are shown in Tab. 4. The experimental results demonstrate that our method outperforms other approaches in terms of both visual quality and lip-sync accuracy of the generated videos. This finding further corroborates that our 3D mesh achieves the highest alignment precision with the ground truth data.

Furthermore, as shown in Tab. 2, the optimal FD scores for both the speaker and listener segments indicate that our method demonstrates superior realism in expression, jaw, and pose compared to the baseline methods. Simultaneously, these optimal SID metrics also show that our method achieves richer and more diverse motion patterns. These experiments collectively validate the effectiveness of our diffusion-based 3D motion generation and 2D-level supervision.

### 4.2.2 QUALITATIVE RESULTS

To intuitively demonstrate the accuracy and naturalness of our method's results, we conduct qualitative comparisons with SOTA methods, as shown in Fig. 5. We present results for both "listening" and "speaking" states. It can be observed that CodeTalker Xing et al. (2023) captures some mouth movements, but the amplitude of mouth changes is small; in speaking scenarios

Table 4: Quantitative comparisons of the SOTA methods on 2D image generation in our MANGO-Dialog testsets.

| Methods | Visual Quality | | | Lip Sync | |
|---|---|---|---|---|---|
| | PSNR↑ | SSIM↑ | LPIPS↓ | LSE-C↑ | LSE-D↓ |
| ARTalk | 25.43 | 0.852 | 0.189 | 4.923 | 7.632 |
| SadTalker | 26.12 | 0.863 | 0.174 | 5.136 | 7.427 |
| AniTalker | 24.87 | 0.832 | 0.241 | 3.279 | 9.362 |
| **Ours** | **26.36** | **0.874** | **0.167** | **5.382** | **7.296** |

where the mouth should be open, it sometimes remains closed (e.g., column 3, row 1 and 4), and in listening states, the mouth sometimes fails to close (e.g., column 9, row 4). DiffPoseTalk Sun et al. (2024) exhibits larger motion amplitudes, but still shows inconsistencies between mouth open-

Table 3: Ablation study on our MANGO-Dialog, including 3D-mesh and 2D-image metrics.

| Version | 3D-mesh Metrics | | | | | | 2D-image Metrics | | | |
|---|---|---|---|---|---|---|---|---|---|---|
| | MVE↓ | LVE↓ | FDD↓ | MOD↓ | MTM↓ | SLCC↑ | PSNR↑ | SSIM↑ | LPIPS↓ | MAE↓ |
| baseline | 1.542 | 0.269 | 1.607 | 1.391 | 4.781 | 0.645 | 18.97 | 0.745 | 0.352 | 0.076 |
| +audio fusion | 1.401 | 0.193 | 1.603 | 1.281 | 4.242 | 0.661 | 20.17 | 0.783 | 0.278 | 0.067 |
| +audio res | 1.454 | 0.233 | 1.601 | 1.230 | 4.371 | 0.683 | 20.53 | 0.789 | 0.257 | 0.063 |
| +indicator | 1.513 | 0.235 | 1.601 | 1.247 | 4.526 | 0.698 | 20.42 | 0.782 | 0.238 | 0.064 |
| +jaw pose | 1.507 | 0.235 | **1.579** | 1.221 | 4.147 | **0.802** | 21.20 | 0.794 | 0.246 | 0.061 |
| Ours(+two stage) | **1.225** | **0.174** | 1.593 | **1.096** | **4.015** | 0.791 | **23.25** | **0.821** | **0.213** | **0.054** |

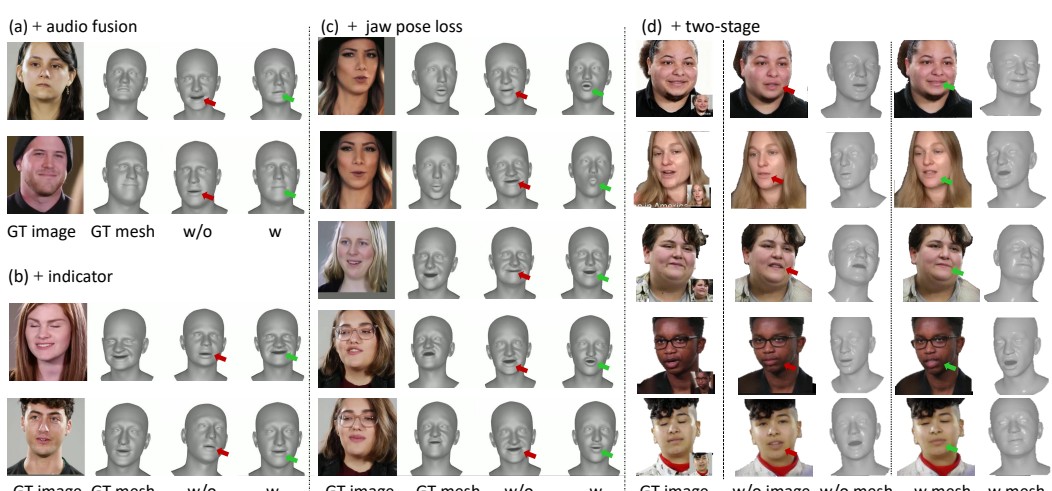

Figure 7: Visual results of ablation study in the 3D-level strategy in stage1 (**a,b,c**) and the two-stage joint training strategy (**d**).

ing/closing and the ground truth (e.g., column 4, row 2 and 3). Compared to DualTalk Peng et al. (2025), although it produces richer mouth dynamics, it sometimes generates overly open mouths (e.g., column 5, row 4), and in closed-mouth scenarios, the mouth remains open (e.g., column 10, row 2, 3, and 4). Notably, compared to the pseudo ground-truth mesh obtained by tracking, our generated meshes sometimes align better with the ground-truth images. For example, the pseudo ground-truth mesh may show excessive mouth opening or incomplete closure (e.g., column 2, row 2 and column 8, row 2), while our mesh does not have these issues. This benefit comes from our joint two-stage training, which achieves 2D-lifted alignment for the 3D mesh.

Meanwhile, we present a comparison of the 2D effects of our method with existing SOTA methods, as illustrated in Fig. 6. It is clearly evident that our 2D results outperform the comparative methods in terms of motion accuracy, synchronization, and identity preservation when compared to the ground truth. More notably, our approach is capable of capturing subtle dynamics in conversations, such as producing a natural smile in conversation (e.g. column 4, raw 4 and column 8, raw 5), without appearing rigid or unnatural.

To further validate these observations, we conducted a user study in which we invited 15 participants to rate the realism and expressiveness of the generated animations. Utilizing the Mean Opinion Score (MOS) protocol, participants rated the realism of the test set's listening and speaking segments across four dimensions: pose naturalness, expression richness, visual quality, and audio-lip synchronization Peng et al. (2025). Additionally, we asked users to evaluate the correlation between the speaker's speech and the listening behavior across three dimensions: temporal synchronization, action appropriateness, and head pose naturalness. The results, presented in Tab. 5, indicate that our method achieved higher average scores across virtually all dimensions compared to the baseline methods.

### 4.3 ABLATION STUDY

**Effectiveness of the DIM.** To validate the effectiveness of the audio fusion module, we conducted an ablation study using DiffPoseTalk Sun et al. (2024) as the baseline. As shown in Fig. 7 **(a)**, the

Table 5: User study results evaluating animation realism, expressiveness, and interaction correlation. Rating is on a scale of 1-5; the higher the better.

| Methods | Realism and Expressiveness | | | | | | | Interaction Correlation | | |
|---|---|---|---|---|---|---|---|---|---|---|
| | L-Visual Quality ↑ | L-Expression Richness ↑ | L-Pose Naturalness ↑ | S-Lip Sync Accuracy ↑ | S-Visual Quality ↑ | S-Expression Richness ↑ | S-Pose Naturalness ↑ | Temporal Coherence ↑ | Contextual Appropriateness ↑ | Pose Naturalness ↑ |
| CodeTalker | 2.6 | 2.6 | 2.8 | 2.1 | 1.8 | 2.2 | 1.3 | 1.2 | 1.5 | 2.8 |
| DiffPoseTalk | 2.3 | 3.4 | 3.5 | **4.4** | 3.8 | 3.8 | 3.9 | 1.7 | 2.1 | 3.5 |
| DualTalk | 3.5 | 3.8 | 3.2 | 4.0 | 3.5 | 3.6 | 3.7 | 2.9 | 3.0 | 3.2 |
| Mango(Ours) | **3.9** | **3.9** | **4.0** | 4.3 | **4.1** | **3.9** | **4.0** | **3.7** | **3.3** | **3.7** |

audio fusion module offers two main advantages: (1) it effectively distinguishes between the speaker and the listener, preventing mouth movements associated with speaking from appearing during listening states (see the upper row of Fig. 7 (**a**)); (2) it establishes a correlation between the speaker and listener's audio, such that when the other person is speaking with a smile, the model also exhibits a corresponding smile (see the lower row of Fig. 7 (**a**)). As shown in Tab. 3, incorporating the audio fusion module leads to significant improvements in both 3D and 2D metrics.

**Effectiveness of Speaking Indicator.** Our experimental results show that incorporating the speaking indicator leads to richer facial expressions and a significant improvement in overall perceptual quality (as shown in Fig. 7 (**b**)). We attribute this to the indicator's ability to help the model accurately distinguish between the speaker and the listener, enabling more targeted interaction modeling.

**Effect of the Second Stage.** Introducing the second stage, as in the last row of Tab. 3, all metrics improve significantly, which fully demonstrates the positive impact of the second stage on the first stage. The notable improvement in 2D metrics indicates that the motion after passing through the 3D GS Renderer becomes more accurate, validating our hypothesis that 2D loss can influence 3D performance, thereby enhancing the accuracy of 3D motion and making the rendered 2D images closer to the ground truth. As shown in Fig. 7 (**d**), before introducing the second stage, many motion sequences and their rendered images cannot be perfectly aligned with the target, especially for mouth dynamics; after introducing the second stage, both the motion and rendered images are well aligned with the target image. This demonstrates that relying solely on 3D pseudo-loss constraints is far from sufficient, and 2D supervision provides a more accurate optimization for motion learning. More ablation study results are shown in C.

To further validate the motivation of employing our stage-two approach and demonstrate the superior accuracy of our mesh generation compared to Spectre under certain specific conditions, we conducted a targeted evaluation. For several selected videos with manually annotated ground-truth keypoints ($X_{gt}$), we first obtained 2D keypoints ($X_s$ and $X_m$) via projection from Spectre's reconstructed meshes and our model's generated meshes, respectively. We then computed the Lip Distance Metric ($D_{Lip}$) and the final Mean Absolute Error ($MAE_{Lip}$) across the test sequences. The detailed calculation procedures for $D_{Lip}$ and $MAE_{Lip}$ are provided in D. The baseline Spectre-Reconstruction method yielded an $MAE_{Lip}$ of $9.68$. In contrast, our method, achieved a significantly lower $MAE_{Lip}$ of $6.23$. The results decisively confirm our hypothesis: Mango demonstrates superior accuracy over Spectre in specific instances, particularly when the Spectre reconstruction includes overly exaggerated or non-physical mouth shapes, leading to a substantial reduction in lip synchronization error. To more intuitively demonstrate (or support) our conclusion, we provide visual comparisons in D.

## 5 CONCLUSION

In this paper, we present MANGO, a method for multi-speaker 3D talking head generation. MANGO fully explores the mapping relationship between speech semantics and lip movements by introducing a diffusion-based dual-audio fusion module. In addition, we incorporate a 3D Gaussian renderer to synthesize images under the supervision of real 2D images. Through a two-stage joint training strategy, our method achieves 2D-aligned 3D mesh generation. We conducted extensive experiments on our self-built dual-speaker conversation dataset, and the results show that our method outperforms existing approaches in both 3D mesh accuracy and 2D image quality. Despite these advances, our method still shows minor artifacts in some non-facial areas. We leave these improvements to future work.

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

## A NETWORK AND IMPLEMENTATION DETALIS

### A.1 DIM ARCHITECTURE.

We feed two speech inputs into Hubert to extract features, obtaining two audio features, $H_{self}$ and $H_{other}$, with a feature dimension of $d = 768$, These features are then projected to $d = 256$, through a linear layer. We concatenate these two features along the feature dimension, resulting in a joint feature $H_{dual}$ with $d = 512$, Next, this feature is passed into a two-layer, 8-head TransformerEncoderLayer and combined with $H_{self}$ using a residual connection, producing an interaction feature that remains at $d = 1024$. Finally, an indicator is appended to the end of the feature dimension, yielding the final fused feature $H_{fuse}$ with a dimension of $d = 513$.

### A.2 FMM ARCHITECTURE.

In our experiments, we set the window sizes to $w_p = 10$, $w = 100$, Our diffusion-based Transformer decoder consists of 8 stacked TransformerDecoderLayer blocks. The total number of diffusion sampling steps is set to 500. A critical architectural component is the encoder-decoder alignment mask, enforcing temporal correspondence between speech and motion representations. Formally, each motion feature is constrained to attend only to its contemporaneous speech feature.

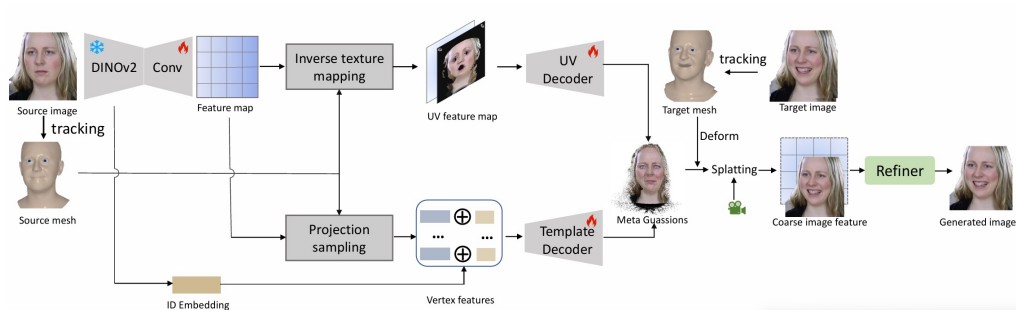

Figure 8: The detailed architecture of our MG-Renderer.

### A.3 META GUASSIANS.

In this section, we model the face in a canonical space using 3D Gaussians. Each Gaussian is defined by its position, rotation, scale, opacity, and a latent appearance vector: $G = \{\mu, r, s, \alpha, c\}$. The meta Guassians are constructed from two components: **(1)** Template Gaussians derived from the FLAME model $G_T = \{\mu_t, r_t, s_t, \alpha_t, c_t\}$, which are relatively sparse and responsible for representing the overall texture and geometry; and **(2)** UV Gaussians attached to the triangulated mesh $G_{UV} = \{\mu_{uv}, r_{uv}, s_{uv}, \alpha_{uv}, c_{uv}\}$, which are used to encode fine-grained details. The detailed architecture of our MG-Renderer is shown in Fig.8.

For the template Gaussians, we first extract the vertex set $V = \text{FLAME}(P_r)$ from the FLAME parameters of the reference image $I_r$ to initialize their mean positions. The vertices $V$ are then projected onto the 2D image plane, and the corresponding features are sampled from the DINOv2 feature map $F_r = \text{DINO}(I_r)$ to facilitate the subsequent decoding of additional Gaussian attributes. This process can be formulated as follows::

$$f_s^i = \mathcal{S}(\mathcal{P}(v_r^i, RT_r), F_r), \tag{11}$$

where $\mathcal{S}$ and $\mathcal{P}$ represent the sampling and projection operators, respectively. We obtain a global embedding $f_{id}$ from the reference image to represent identity, and an optimizable base feature $f_b^i$ to capture specific semantic information. These three features are concatenated and passed through a hierarchical MLP-based decoder $D$. Finally, we concatenate these features to obtain the attribute vector for each Gaussian point: $\{r_t^i, s_t^i, \alpha_t^i, c_t^i\} = D_v(f_s^i \oplus f_b^i \oplus f_{id})$.

Relying solely on template Gaussians to represent the avatar fails to capture high-frequency details due to the limited number of Gaussians, and it also struggles to generalize to regions not well covered by the template. To address this limitation, we construct UV Gaussians based on a UV texture map. Specifically, each triangle on the mesh corresponds to a UV Gaussian, whose mean position $\mu_{uv}$ is defined as the barycenter of the triangle. The UV Gaussian centers are then projected onto the DINOv2 feature map $F_r$, from which the corresponding features are sampled to form the UV feature map $F_{uv}$. CNN decoder $D_{uv}$ are subsequently employed to decode various UV Gaussian attributes $\{\Delta\mu_{uv}, r_{uv}, s_{uv}, \alpha_{uv}, c_{uv}\} = \mathcal{D}_{uv}(F_{uv})$, resulting in the final UV Gaussian representation denoted as $G_{UV} = \{\Delta\mu_{uv} + \mu_{uv}, r_{uv}, s_{uv}, \alpha_{uv}, c_{uv}\}$, which is used for rendering.

### A.4 LOSS WEIGHTS

The loss weights of our stage-1 are as followings: $\mathcal{L}_{jaw} = 0.2$, $\mathcal{L}_{vert} = 2e6$, $\mathcal{L}_{vel} = 1e7$, $\mathcal{L}_{smooth} = 1e4$. The loss weights of our stage-2 are $\lambda_{pho} = 1.0$, $\lambda_{per} = 0.025$.

### A.5 EVALUATION METRIX

In our experiments, we mainly evaluate our method from two aspects: 3D mesh modeling and 2D image quality, to comprehensively assess the performance of our approach in dual-speaker dialogue scenarios. For 3D mesh modeling, we use key metrics such as lip vertex error (**LVE**) Richard et al. (2021a) and mean vertex error (**MVE**) to measure the frame-wise differences between mouth vertices and overall facial vertices compared to tracking results, use upper face dynamics deviation

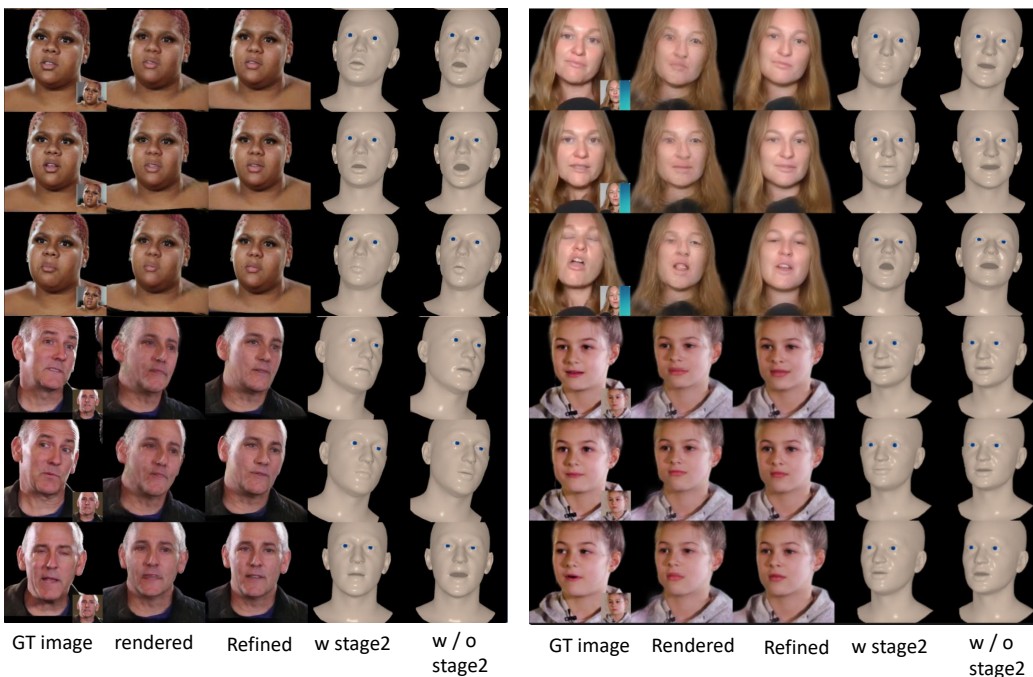

GT image   rendered   Refined   w stage2   w / o stage2        GT image   Rendered   Refined   w stage2   w / o stage2

Figure 9: More visual results of ablation study on the two-stage joint training strategy

(**FDD**) Xing et al. (2023) to evaluate the accuracy of inter-frame vertex motion and use **MOD** to represent the speech-synchronized fidelity of lip opening/closing movements. Notably, we also employ the recently proposed metrics from Chae-Yeon et al. (2025): Mean Temporal Misalignment (**MTM**) and Speech and Lip Intensity Correlation Coefficient (**SLCC**), which are used to evaluate the temporal consistency of 3D meshes and the correlation between lip and speech intensity, respectively. For 2D image, we use Peak Signal-to-Noise Ratio (**PSNR**), Structural Similarity Index Measure (**SSIM**), Learned Perceptual Image Patch Similarity (**LPIPS**) Zhang et al. (2018) to assess the image-level generation quality. For image-level evaluation of lip synchronization and mouth shape, we adopt perceptual metrics from Wav2Lip Liang et al. (2024), including the distance score (**LSE-D**) and confidence score (**LSE-C**).

### A.6   TRAINING DETAILS

The sequence length for the first stage is set to $w = 100$, and in the second stage, we randomly select $n = 5$ frames. We first pretrain the two stages separately, both using a single NVIDIA RTX 3090 GPU and the Adam optimizer, with batch sizes of 16 and 6, respectively.The initial learning rate is set to $1 \times 10^{-4}$ for both. Afterwards, we use a single A6000 GPU to jointly train both stages, with a batch size of 2 and a training time of about 12 hours. The learning rates for two stages are respectively scheduled using the Warmup Scheduler and Decay Scheduler, with 10k and 200k iterations, and training times of 4 hours and 5 days, respectively. Afterwards, we use a single A6000 GPU to jointly train both stages, where the first stage adopts a cosine annealing learning rate to help the model converge more efficiently, while the second stage maintains the same settings as in pretraining.

### B   DATASET DETAILS

Our video dataset was sourced from diverse YouTube channels, encompassing a wide variety of scenarios. Using scene detection and segmentation algorithms, we extracted two-person conversation clips within consistent scenes. Specifically, our pipeline for processing conversational videos consists of the following steps: First, we employ TalkNet Tao et al. (2021) to detect and separate the audio-synchronized speech segments and corresponding visual frames for both speakers. All single-

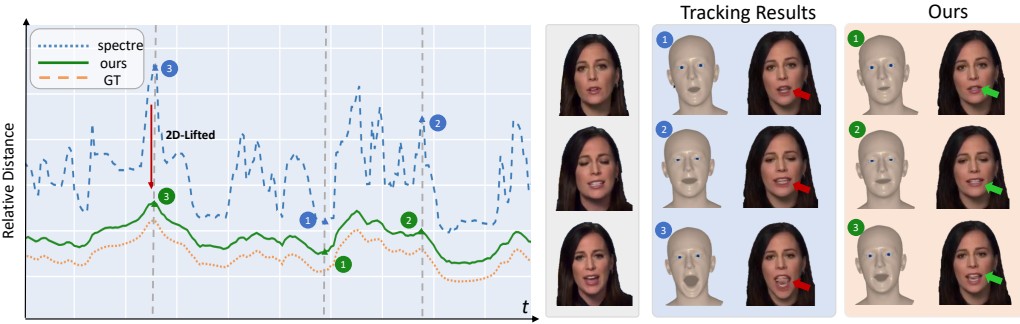

Figure 10: Analysis of our method. The left side shows the curve comparing the relative average mouth distance to the ground truth over a video sequence for mainstream tracking methods, our method, and the ground truth itself. The right side provides a visualization of the meshes and their corresponding rendered images for three selected frames. It can be observed that our method demonstrates superior alignment with the ground truth, both in terms of the curve trajectory and the individual frame meshes and images.

person frames are in 1920×1080 resolution recorded at 25 frames per second, with audio sampled at 16kHz.

This process also extracts their active speaking intervals, which serve as speaking indicators for subsequent analysis. Upon obtaining the processed videos, we utilize the 3D reconstruction method Spectre Filntisis et al. (2022) to track FLAME motion parameters, followed by a keypoint-based alignment optimization to refine the camera parameters.

Finally, we divided all clips into three parts: the training, validation, and test sets, containing 4,300 clips, 400 clips, and 200 clips, respectively. To evaluate the model's generalization ability, the identities in the testset were unseen during training. We used the testset as the dataset for subsequent metric evaluation.

## C MORE ABLATION STUDY RESULT

**Effect of Jaw-pose Loss.** As shown in Tab. 3, adding jaw pose loss significantly improves the SLCC and MTM metrics, indicating that the generated motion sequences are more correlated with the audio. This is also visually demonstrated in Fig. 7 **(c)**, where jaw pose loss leads to richer and more realistic mouth dynamics, especially for pronounced actions such as pouting, making the generated mouth movements closer to those in real speech.

**Effect of the Second Stage.** To further demonstrate the effectiveness of our proposed two-stage framework and the joint training of both stages, we present additional case studies here. The results are shown in the Fig. 9, where from left to right are: ground-truth image, rendered image, refined rendered image, mesh without stage-2, and mesh after stage1-stage2 joint training. It can be observed that after incorporating the second stage, the mesh aligns significantly better with the ground-truth image, particularly in terms of mouth opening/closing degree. Without the second stage, many cases that should have closed mouths fail to do so.

**Robustness Test for the Indicator.** We adopted consecutive misattribution noise to simulate misattribution errors in real-world scenarios. Specifically, given a segment of speech with length $L$, we obtain the noisy mask $\tilde{A}'$ by performing an XOR operation ($\oplus$) between the original speaker mask $A$ and a consecutive noise mask $\Delta A'$:

$$\tilde{A}' = A \oplus \Delta A'$$

where $\Delta A'$ is generated by randomly selecting a consecutive segment of length $L_{\text{flip}}$ in $A$ and flipping its 0/1 values. The length $L_{\text{flip}}$ is defined as:

$$L_{\text{flip}} = \lceil \alpha \cdot L \rceil, \quad \text{with } \alpha \in [0.05, 1]$$

**(1)** Our experimental results show that the model's performance did not degrade significantly even with an error rate where the segment length factor $\alpha$ reached 0.3 (i.e., 30% of the consecutive segments were flipped), which demonstrates the strong robustness of our model. This anti-interference

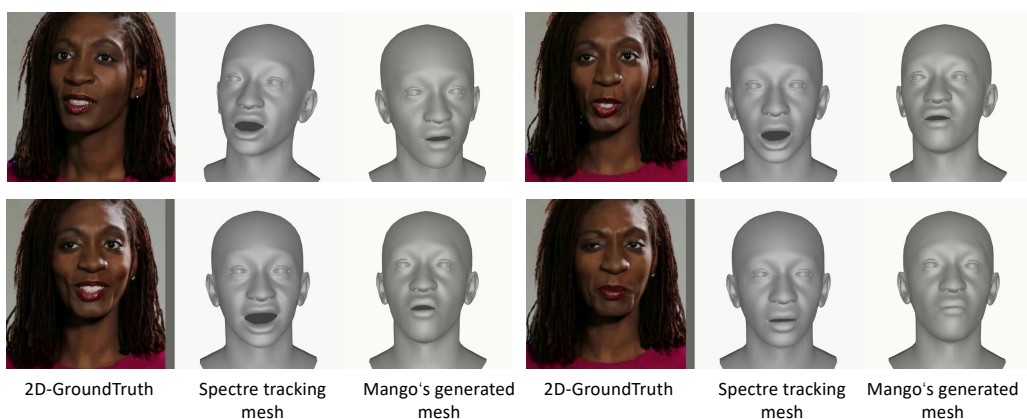

|2D-GroundTruth | Spectre tracking mesh | Mango's generated mesh | 2D-GroundTruth | Spectre tracking mesh | Mango's generated mesh |

Figure 11: A comparison of the Spectre tracking mesh and our generated mesh against the 2D ground truth image reveals that the Spectre mesh exhibits an overly exaggerated mouth opening, whereas our mesh is visually better aligned with the 2D ground truth image.

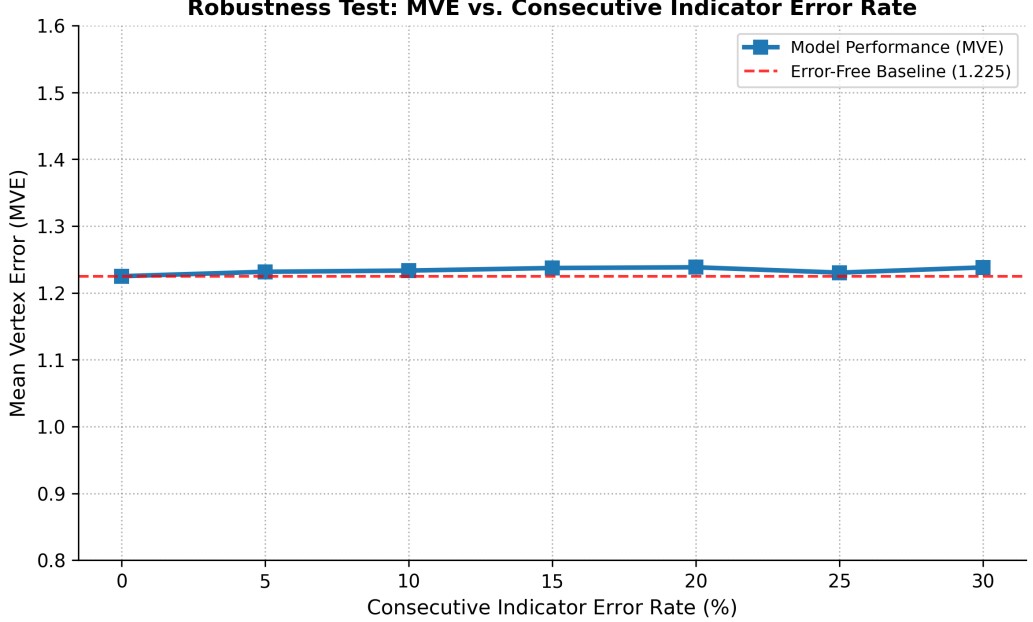

Figure 12: **Robustness Test against Indicator Misattribution Errors.** Model performance (MVE) under varying Consecutive Indicator Error Rates ($\alpha$). The curve demonstrates strong robustness, with minimal performance degradation up to $\alpha = 0.3$.

capability primarily stems from the model's reliance on longer-term speech features to infer the dialogue's logic and context, which effectively mitigates the negative impact caused by short-term, consecutive 0/1 misattribution errors.

**(2)** we observe that the indicator prediction accuracy of the speech separation model we utilize exceeds $70\%$ (specifically, $90.8\%$ Tao et al. (2021)), which suggests our model is robust in most scenarios. We present the curve illustrating the change in model performance as the consecutive flip segment length factor $\alpha$ varies in Fig.12.

## D  EFFECTIVENESS ANALYSIS

To accurately assess the performance of 3D reconstruction and generation methods, we utilize the Lip Distance Metric ($D_{\text{Lip}}$) to quantify the accuracy of mouth movements. This metric first involves projecting the reconstructed 3D vertices onto the 2D space via camera projection to obtain the corresponding 2D planar keypoints, $X_m$ and $X_s$. Ground-Truth keypoints $X_{gt}$ are obtained

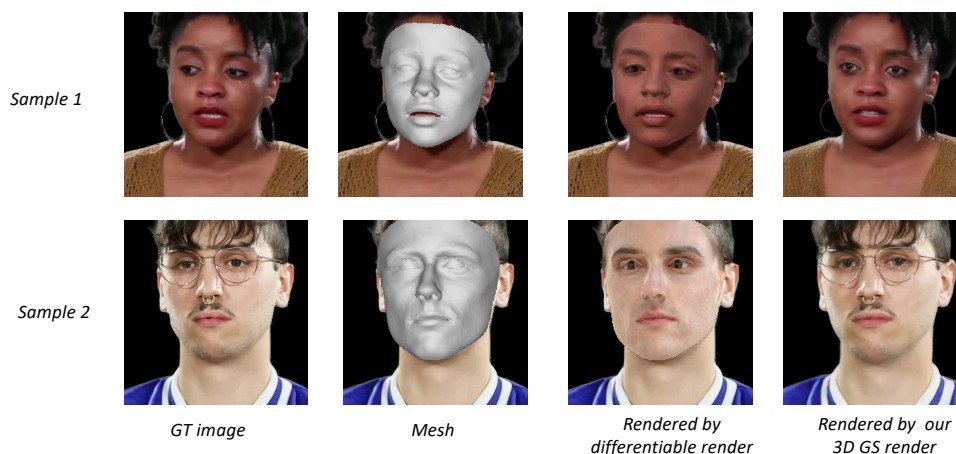

*Sample 1*

*Sample 2*

GT image    Mesh    Rendered by    Rendered by our
differentiable render    3D GS render

Figure 13: **Comparsion of differentiable render with our 3DGS Renderer**. Clearly, while both methods preserve the mesh's geometric structure, our 3D GS Renderer yields significantly higher fidelity images compared to the differentiable renderer.

by manually annotating the corresponding GT images. The $D_{\text{Lip}}$ is calculated statistically as the average Euclidean distance between the corresponding keypoint pairs of the upper and lower lips for each frame, serving as a measure of the degree of mouth opening and closing. The calculation formula for the Lip Distance Metric $D_{\text{Lip},t}$ is as follows:

$$D_{\text{Lip},t} = \frac{1}{N} \sum_{i=1}^{N} \sqrt{(x_{i,t}^U - x_{i,t}^L)^2 + (y_{i,t}^U - y_{i,t}^L)^2}$$

where $x_{i,t}^U, y_{i,t}^U$ and $x_{i,t}^L, y_{i,t}^L$ represent the x-coordinate and y-coordinate of the $i$-th pair of upper and lower lip keypoints, respectively, at time $t$. We calculated this metric for the three sets of keypoints (Mango, Spectre, and GT) to obtain $D_{\text{Lip},t}^{\text{mango}}$, $D_{\text{Lip},t}^{\text{spectre}}$, and $D_{\text{Lip},t}^{\text{gt}}$, which are used to generate the curves in Fig.10 (a sampled video case in our manually annotated video). Finally, we statistically calculated the Mean Absolute Error (MAE) across the test sequences to evaluate our method (Mango-Generation) against the Spectre-Reconstruction approach.The MAE for the Spectre-Reconstruction was 9.68, while the MAE for the Mango-Generation (our method) was 6.23. Fig.10 presents an effectiveness analysis of our method. The curve on the left demonstrates that our method produces a mesh significantly closer to the ground-truth image. Its trend indicates superior temporal consistency and stability over Spectre, while the amplitude reveals higher per-frame accuracy. On the right, we compare the ground-truth image with: the mesh after tracking, its rendered image, our method's mesh, and our corresponding rendered image. A visual comparison clearly shows that the mouth opening amplitude of the tracking-based method deviates from the ground truth, whereas our result nearly matches it perfectly, achieving a 2d-lifted effect. To provide a more intuitive illustration, we directly present a visual comparison of the Spectre tracking mesh and our generated mesh against the 2D ground-truth image in the Fig.11.

**Why use the 3DGS renderer instead of a differentiable renderer?** Although most 3D face reconstruction methods use a single image as a supervision signal, we are the first to propose its use in the audio-driven motion sequence generation task. Furthermore, the 3D face reconstruction field predominantly utilizes differentiable renderers, but this approach has limitations: optimizing parameters like 3D geometry, camera, albedo, and illumination is inherently an ill-posed problem [2]; simultaneously, the domain gap between the differentiable rendered image and the real image severely hinders effective model learning Retsinas et al. (2024); Liu et al. (2025), leading to insufficient realism in the rendering results. Here, we innovatively propose the use of a 3D GS renderer. Compared to traditional differentiable renderers, the gap between the 3D GS rendered image and the real image is significantly reduced, greatly alleviating gradient instability. Compared to differentiable renderers[4], the 3DGS renderer offers a significant overall advantage in visual fidelity, greatly narrowing the gap between the rendered image and the real image. The comparison of 3DGS and differentiable renderers is detailed in Fig.13.

# E    ETHICS CONSIDERATIONS

The development of MANGO raises a series of significant ethical issues that require in-depth exploration, especially concerning privacy protection, potential misuse, and broader societal impacts. Although the dataset is constructed from publicly available conversational data across multiple online platforms, we have implemented strict data anonymization measures and complied with current data privacy regulations to protect user privacy throughout the data collection and processing stages. However, as technology continues to advance, we recognize the need to continuously strengthen these protective measures to prevent the unintentional leakage of sensitive personal information.

A particularly pressing ethical concern is the risk of malicious use. The advanced conversational generation capabilities of MANGO could be exploited, leading to misleading synthetic dialogues or unauthorized impersonations. To address these potential threats, we are developing technical solutions, such as digital watermarking systems, to accurately identify AI-generated content. Additionally, we are formulating clear usage policies and ethical guidelines to regulate the use and access to this technology. These measures will be integral components of any future public release, aiming to effectively prevent improper applications while promoting responsible technological innovation.

The ethical framework surrounding MANGO will also evolve alongside technological advancements, requiring close collaboration with ethicists, policymakers, and the broader AI community to develop appropriate governance structures and usage standards for this emerging technology. Only in this way can we ensure that technological progress is made while safeguarding the overall interests and security of society.

# F    LIMITATIONS AND FUTURE WORKS

Although our approach can generate natural 2D-3D conversational digital humans, there are still two main shortcomings: first, we are unable to capture the subtle details of expressions and movements during conversations, and emotions may be misrepresented; second, when there are significant head movements, frame distortion occurs. In the future, we will collect more datasets with complex emotions to enhance the model's generalization capabilities. Additionally, during the rendering phase, we will provide the model with more prior information to ensure that it can still generate realistic images despite large pose variations. Furthermore, due to the limited accuracy of voice separation methods, it is challenging to separate voices when both parties are speaking simultaneously, which hinders the model's learning efficiency. We are also seeking better methods to improve the model's learning effectiveness. On the other hand, the use of a diffusion model in our first stage introduces some computational overhead, increasing the inference time. In the future, we will dedicate our efforts to leveraging diffusion acceleration techniques to improve the inference speed.

# G    USE OF LLMS AND REPRODUCIBILITY STATEMENT

**Usage of LLMs.** We only used LLMs as a language polishing tool, without involving them in method design, experimental design, or any other aspects.

**Reproducibility.** We ensure that our method is fully reproducible, and we will publicly release the training data, code, and model weights upon paper acceptance.

