# OpenReview forum: "MANGO: Natural Multi-speaker 3D Talking Head Generation via 2D-Lifted Enhancement"
_ICLR.cc/2026/Conference — Submitted to ICLR 2026_

### Official Review · Reviewer_X7SN · 2025-10-25

**Soundness:** 3
**Presentation:** 3
**Contribution:** 3
**Rating:** 6
**Confidence:** 4

**Summary:**

This paper presents MANGO, a two-stage framework for multi-speaker 3D talking head generation. The first stage uses a diffusion-based dual-audio motion generator to produce 3D motion parameters conditioned on both speakers’ audio. The second stage employs a 3D Gaussian Renderer (MG-Renderer) to synthesize high-fidelity images using 2D photometric supervision, which the authors refer to as a 2D-lifted enhancement. A new dataset, MANGO-Dialog, is also introduced, consisting of over 50 hours of synchronized 2D-3D conversational data from 500+ speakers. Experimental results demonstrate quantitative and qualitative improvements over prior 3D talking head methods such as DualTalk and DiffPoseTalk, particularly in lip-sync precision and overall mesh–image alignment.

**Strengths:**

- Novel Dual-Audio Diffusion Framework : The combination of a dual-audio fusion module and a diffusion-based motion generator allows modeling of bidirectional conversational dynamics, distinguishing speaking and listening states more effectively than single-speaker systems.
- Two-Stage 2D-Lifted Enhancement Strategy : The introduction of 2D image-level supervision through the Gaussian Renderer effectively refines 3D mesh predictions, mitigating the noise of pseudo-3D labels obtained from tracking.
- New Dataset (MANGO-Dialog) : A large-scale, 2D–3D aligned multi-speaker dialogue dataset is a valuable contribution that could benefit future research in multi-person conversational synthesis.
- Comprehensive Evaluation : Extensive comparisons with single- and multi-speaker baselines (FaceFormer, DualTalk, SadTalker, etc.) show improved 3D accuracy (LVE, MVE) and 2D fidelity (PSNR, SSIM, LPIPS).

**Weaknesses:**

- Limited Conceptual Novelty : The framework is primarily a combination of existing paradigms. 3D talking head generation and speech-based motion diffusion—with modest architectural novelty. The overall system resembles a combination of talking head generation and speech separation rather than a fundamentally new paradigm.
- Scalability Concerns : The method generates videos at a head level, which limits scalability to full-scene multi-speaker synthesis. Extending the approach to simultaneous multi-agent scenes or long-turn interactions would be challenging given the per-head rendering design.
- Problem Scope Overlap : The targeted issue of over-smoothed mouth motion is not unique to multi-speaker setups; numerous single-speaker 3D talking head works (e.g., DiffPoseTalk, FaceFormer) already address similar issues with comparable diffusion or transformer-based solutions.
- Relatively Lower Visual Quality : Compared to high-quality 3D generative systems such as HALLO3 or LivePortrait, the generated outputs in Fig. 6 appear less photorealistic and expressive. Leveraging stronger generative priors might substantially improve realism and lip-sync fidelity.
- Terminological Ambiguity.The term “2D-lifted enhancement” is not clearly justified : It appears to describe the process of applying 2D photometric loss to refine 3D motion, but the phrasing could mislead readers into thinking it’s a new geometric transformation rather than a training strategy.

**Questions:**

- Advantage over Task Composition : Since the proposed setup essentially combines speech-driven motion synthesis with conversational context modeling, what specific benefits does MANGO achieve beyond simply combining existing talking head and speech separation modules?
- Scene-Level Generation : Could the method be extended to generate an entire two-speaker video scene simultaneously, rather than per-head? If so, what architectural or computational challenges arise due to the current 3D representation?
- Relation to Single-Speaker 3D Methods : How does the proposed system differ from prior works that already tackled over-smoothing using diffusion or Gaussian renderers (e.g., DiffPoseTalk, ARTalk)? Are the improvements mainly empirical or conceptual?
- Quality Gap vs. MANGO-Dialog Baselines : The paper shows improved quantitative metrics, but the generated samples from MANGO-Dialog still look coarse compared to prior generative 3D methods. Have you considered integrating more advanced generative models like LivePortrait or HALLO3 to enhance appearance realism?
- Clarification on “2D-Lifted Enhancement : ”Could you clearly define this term? Is it equivalent to the two-stage alternated supervision process (3D→2D refinement), or does it imply a structural connection between 2D features and 3D geometry?

**Details Of Ethics Concerns:**

The task involves generating human talking head videos, which inherently raises ethical considerations related to identity manipulation, consent, and potential misuse in deepfake applications. While the paper focuses on technical contributions, responsible research practices—such as data collection transparency, human subject consent, and safeguards against misuse—should be clearly discussed.

---

> ### Author Response · Authors · 2025-11-22
> **Response to X7SN**
>
> Dear Reviewer X7SN,
>
> Many thanks to your valuable comments and questions, which help us a lot to improve our work. We address your questions as follows.
> ***
> > **Q1:** Regarding the novelty of the MANGO network structure: Aside from combining existing talking head and speech separation modules, what specific advantages does MANGO offer?
>
> **A1:** Thank you for raising this point. We want to clarify our novelty and advantages:
> 1. First, we are different from the traditional talking head method. Our input includes the speech of both the speaker and the listener. Our 'talking head' is an interactive talking head that can not only speak but also listen, and naturally transition between speaking and listening states. Traditional talking head methods, in contrast, can only handle single audio-driven scenarios.
>
> 2. Second, simply feeding speech separation output into a talking head model does not yield realistic results when the agent should be in a listening state. As shown in our supplementary material's demo, DiffPoseTalk and CodeTalker still exhibit unnecessary mouth movements ("random movement") while listening, and DualTalk sometimes appears completely static in the listening state. Our model, however, can not only produce a realistic listening outcome but also exhibit appropriate responses, such as smiling, blinking, nodding, and more.
>
> 3. The source of this capability is our novel design: We engineered a Dual-person Audio Interaction Module (DIM) that uses a Transformer to learn the interactive features within the two parties' speech, establishing a correlation between listening and speaking actions. Furthermore, we employ an Indicator to help the model distinguish the agent's current speaking/listening state, enabling it to model both speaking and listening movements and achieve natural state transitions.
>
> 4. To obtain more accurate mouth movements and shapes, we designed a two-stage training strategy. By introducing 2D supervision, we enhance the accuracy of the 3D mesh, effectively breaking through the model ceiling imposed by the inaccuracies of 3D reconstruction methods. This allows for more precise lip modeling, which is highly significant for interactive digital humans.
>
>
> ***
> > **Q2:**  Scalability concerns. Can the method be scaled to generate full two-speaker video scenes simultaneously, rather than just head-by-head? If so, what architectural or computational challenges arise due to the current 3D representation?
>
> **A2:**  Thank you for your question, which points to an extremely meaningful direction for our future work. Theoretically, our method can be extended to full human body generation. Since the SMPLX model [1] exists for the body, in addition to the FLAME model for the head, we can extend the current paradigm of predicting FLAME parameters (head only) to predicting FLAME + SMPLX parameters. However, the lack of datasets for full-body, multi-person dialogue scenarios remains a major challenge, and the modeling of the relationship between speech and body movements will be significantly more complex. We will explore new architectures and data collection paradigms to realize interactive digital human generation for full-body scenes.
>
> - [1] SMPL-X: A new joint 3D model of the human body, face and hands together

---

> > ### Author Response · Authors · 2025-11-22
> > **Response to X7SN**
> >
> > ***
> > > **Q3:**  Over-smoothed lip movement is not unique to multi-speaker settings, as existing single-speaker works (e.g., DiffPoseTalk, FaceFormer) use similar solutions. How does your system conceptually or empirically differ from prior work (e.g., DiffPoseTalk, ARTalk) that used diffusion/Gaussian renderers to solve over-smoothing?
> >
> > **A3:**
> > Thank you for your question. This is quite helpful for highlighting the distinctive features of our paper.
> > 1. Our paper does not primarily address the "over-smoothed lip movement" problem. The over-smoothing issue we mentioned in Figure 2 is our evaluation of the effect of the existing SOTA 3D reconstruction method, Teaser. It serves as evidence for our viewpoint that "current SOTA 3D reconstruction methods are inaccurate." This is precisely why we must introduce 2D supervision to compensate for this bias. Both DiffPoseTalk and FaceFormer still rely on pseudo-labels derived from existing 3D reconstruction methods for supervision, and they are not focused on solving the specific problem we tackle.
> > 2. Our improvement is conceptual. We aim to resolve the limited model performance ceiling caused by the inaccurate motion modeling inherent in Spectre as a supervision model. Thus, we designed a two-stage network where the mesh is projected to a 2D image space via the GS renderer, and we leverage 2D-level image supervision to break this bottleneck. From a high-level perspective, we simultaneously utilize 2D and 3D supervision with the goal of "retaining 3D controllability while achieving 2D alignment." Neither DiffPoseTalk, ARTalk, nor other models are based on this motivation, nor have they proposed our specific approach. While our implementation uses the currently popular combination of diffusion and 3D GS, our architecture and training paradigm can still adapt to newer model frameworks in the future should better ones emerge.
> >
> > > **Q4:** The generated output in Figure 6 appears lacking in photorealism and expressiveness when compared to high-fidelity 3D generation systems like HALLO3 or LivePortrait. Has integrating more advanced generative models like LivePortrait or HALLO3 to enhance visual fidelity been considered?
> >
> > **A4:** Thank you for your question; this is highly insightful.
> > 1. While methods based on diffusion (HALLO3) or implicit motion space (LivePortrait) might yield superior 2D visual results in pursuit of visual quality, our objective is to achieve highly controllable, high-fidelity dialogue-driven 3D mesh generation, not end-to-end single-person 2D Talking Head generation or reenactment, as done by HALLO3 or LivePortrait.
> > 2. Our motion space is fundamentally different from that of LivePortrait or HALLO3, making direct integration of their structures impossible. We learn explicitly decoupled FLAME parameters to obtain 3D point clouds. In contrast, LivePortrait models an implicit motion space which, despite having decoupling capabilities, lacks 3D attributes. HALLO3, on the other hand, operates entirely within the video generation paradigm, directly using a Transformer within the 3D VAE latent space to capture the spatio-temporal relationship between audio embeddings and video frame latent codes, without explicitly modeling motion.
> > 3. Regarding the issue of realism and expressiveness you raised, we have taken this into consideration. While maintaining the 3D geometric structure, the current 3D GS renderer we use already demonstrates more realistic results than traditional differentiable renderers. However, the realism and expressiveness still cannot match warping-based methods (LivePortrait) or Diffusion-based methods (HALLO3). It is worth noting that the renderer we employ offers a significant speed advantage (100+FPS). We hope to meet your requirements by continuing to improve the renderer's capabilities—for example, by changing the current single-image input to a multi-image input to model multi-view 3D GS, thereby enhancing the modeling of skin texture and details.
> >
> > > **Q5:** What specifically does "2D-lifted enhancement" refer to? Is it equivalent to the two-stage alternating supervision process (3D → 2D refinement), or does it imply a structural link between 2D features and 3D geometry?
> >
> > **A5:** Thank you very much for your question. It greatly helps to improve the clarity of our paper. Our term "2D-lifted Enhancement" does not refer to a new geometric transformation method, but rather a clear training strategy and supervision process. Specifically, we project the 3D geometric mesh onto the 2D image space via the renderer, and then use the 2D loss to optimize the 3D mesh, thereby "lifting" the 3D accuracy. We have updated the relevant description in the revised version.

---

### Official Review · Reviewer_sCDj · 2025-10-27

**Soundness:** 2
**Presentation:** 2
**Contribution:** 2
**Rating:** 4
**Confidence:** 3

**Summary:**

This paper introduces MANGO, a 3D conversational multi-speaker talking-head generation framework that unifies the synthesis of both speaking and listening behaviors.
The framework consists of two stages:
1. A diffusion-based multi-audio fusion model that models motion distributions across speakers;
2. A 3D Gaussian Splatting (3DGS) renderer that converts predicted motion sequences into videos, with additional 2D image supervision to mitigate inaccuracies from 3D tracking.
The authors also provide a new 3D conversational dataset for training and evaluation.

**Strengths:**

1. Presents a unified framework for generating both speaking and listening 3D talking heads. It is a novel and ambitious direction.

2. Incorporating 2D image-level loss after 3DGS rendering helps partially alleviate the errors caused by 3DMM estimation, providing additional supervision for the 3D talking head generation.

**Weaknesses:**

1. (Major) Limited effectiveness in both speaking and listening states: From the demo videos, while the model can roughly switch between speaking and listening modes, neither mode performs convincingly.

    - Speaking: The lip motions are not accurate and clearly worse than single-speaker baselines such as CodeTalker or DiffPoseTalk. Even though MANGO separates speaking and listening audio inputs and introduces an indicator for speaking status, the generated lips remain unsatisfactory. This raises the question of whether the dual-audio module introduces interference between the two states.
    - Listening: The listening behaviors appear almost random or static, lacking clear correlation with the interlocutor’s speech (e.g., at 00:38 in the demo, when hearing “luckily”, DualTalk shows a smile but MANGO does not). The dataset examples contain rich listening behaviors — nodding, smiling, eyebrow raises, or thoughtful blinking — yet these are not reflected in the results. Quantitative or qualitative evidence showing the correlation between listening behavior and input audio would strengthen the claim.
    - In conclusion, the framework currently fails to convincingly capture both accurate lip articulation and expressive listening dynamics.

2. (Major) Questionable benefit of 2D image loss after 3D reconstruction: While introducing a 2D image loss after rendering is presented as a core contribution, such image-space supervision has long been standard in 3D face reconstruction pipelines.
Here, applying the 2D loss after generation introduces compounded errors — both from inaccurate expression estimation and imperfect 3D rendering.
The actual effectiveness of this design is unclear and requires visual ablation evidence.
Moreover, the rendered frames in the demo show strong 3DGS artifacts, raising doubts about gradient stability and potential negative impacts on 3DMM coefficient learning.
As an alternative, will it be more stable and effective to optimize the predicted 3DMM (pGT) directly through differentiable rendering and computing the image loss on the rendered image of pGT?

3. (Major) Relation to INFP remains underspecified: Although the authors claim their task differs from INFP (which generates 2D talking heads), MANGO ultimately renders to 2D and mainly differs in that it uses 3DMM as the intermediate representation instead of INFP’s motion features.

    - The two tasks and formulations are thus highly similar, and a visual comparison with INFP would be essential to demonstrate advantages in motion controllability.

    - Both methods employ a dual-audio module to link speech and motion features. What's the difference and strength of MANGO's audio2motion model against the INFP's?

4. (Minor) Presentation and clarity issues:
    - The two claimed contributions, conversational talking-head generation and 2D image loss, appear weakly connected and seem like two independent ideas. And the statement “in conversational scenarios, lip movements become more complex due to the dynamics of interaction” (L80) lacks empirical justification; single-speaker data can exhibit similar complexity.
    - The naming of Stage 1/2 and Training Phase 1/2 is confusing. For example, does Training Phase 1 (Stage 2 training) refer to only training the MG-Renderer on pGT meshes with 2D image loss?
    - In Equation (3), both $H_{self}$ and $H_{other}$  pass through the Transformer jointly, which seems inconsistent with the schematic in Figure 4.

**Questions:**

1. In our understanding, a listener’s expressions during conversation should depend not only on the other party’s speech content but also on the speaker’s facial expressions. Will the speaking state (or visual features of the speaker) be considered as an additional input when modeling the listening behavior?

---

> ### Author Response · Authors · 2025-11-22
> **Response to sCDj**
>
> Dear Reviewer sCDj,
>
> Many thanks to your valuable comments and questions, which help us a lot to improve our work. We address your questions as follows.
> ***
> > **Q1:** The framework currently fails to convincingly capture both accurate lip articulation and expressive listening dynamics simultaneously. Lip movements during speaking are less accurate than CodeTalker or DiffPoseTalk, and does the dual-audio module introduce interference between the two states? Listening behavior lacks explicit association with the interlocutor’s speech. Provide quantitative or qualitative evidence to demonstrate the correlation between listening behavior and the input audio.
>
> **A1:** Thank you for your question, which is of great significance to our paper.
>
> 1. Our 3D metrics in Table 1 (LVE and MVE measuring motion accuracy, MTM measuring audio-lip synchronization, and SLCC measuring motion-speech correlation) already demonstrate that our method is superior to CodeTalker or DiffPoseTalk, and these metrics are specifically used to measure the accuracy of lip movements.
>
> 2. Regarding the point you raised about DualTalk showing a smile at 00:38 in the demo, this specific expression persists throughout DualTalk's entire listening state in that video; it is not triggered by hearing the word "luckily." Furthermore, DualTalk's expression appears somewhat dull. In comparison, our head poses are more natural.
>
> 3. The dual-audio module did not interfere with the generation of these two states. On the contrary, it captured the interaction information between the two speakers, making the speaker turn-taking transition more natural. Specifically, you can refer to the ablation results of our DIM module in the **demo_sCDj.mp4** (1min23s-2min0s).
>
> 4. Our generated listening behaviors also include noticeable nodding and smiling actions, as detailed in our **demo_sCDj.mp4 (0-36s)**. Concurrently, we conducted a diversity analysis of our generated listener behavior using the quantitative SI Diversity (SID) metric. This metric quantifies diversity by applying k-means clustering to the motion sequences in the feature space and calculating the entropy of the cluster assignment histogram. A higher SID value indicates richer and more diverse motion patterns in the generated animation. The results in the table below demonstrate that our model achieves the best diversity across pose and expression (exp).
>
>     | Method | SID-pose $\uparrow$ | SID-exp $\uparrow$ |
>     | :--- | :--- | :--- |
>     | DiffPoseTalk | 1.03 | 1.16 |
>     | DualTalk | 1.06 | 1.18 |
>     | **Ours** | **1.17** | **1.32** |
>
> 5. We invited 15 participants to use the Mean Opinion Score (MOS) protocol to rate the correlation between the speaker's speech and the listening behavior [1], with a maximum score of 5, across three dimensions: temporal synchronization, action appropriateness, and head pose naturalness. The results indicate that our method achieved higher average scores across all dimensions.
>
>     | Methods | Temporal Coherence $\uparrow$ | Contextual Appropriateness $\uparrow$ | Pose Naturalness $\uparrow$ |
>     | :--- | :---: | :---: | :---: |
>     | CodeTalker | 1.2 | 1.5 | 2.8 |
>     | DiffPoseTalk | 1.7 | 2.1 | 3.5 |
>     | DualTalk | 2.9 | 3.0 | 3.2 |
>     | **Mango(Ours)** | **3.7** | **3.3** | **4.0** |
>
> - [1] DualTalk: Dual-Speaker Interaction for 3D Talking Head Conversations

---

> ### Author Response · Authors · 2025-11-22
> **Response to sCDj**
>
> > **Q2:**  Image space supervision is standard practice in the 3D face reconstruction field, and the practical effectiveness of this design is unclear. Furthermore, 3DGS artifacts may affect parameter learning and the stability of gradients during model training. Could image-level loss be calculated via a differentiable renderer?
>
> **A2:**
> 1. Although most 3D face reconstruction methods use a single image as a supervision signal, we are the first to propose its use in the audio-driven motion sequence generation task, a concept that Reviewer sYZJ and Reviewer X7SN both agreed upon. Furthermore, the 3D face reconstruction field predominantly utilizes differentiable renderers, but this approach has limitations: optimizing parameters like 3D geometry, camera, albedo, and illumination is inherently an ill-posed problem [2]; simultaneously, the domain gap between the differentiable rendered image and the real image severely hinders effective model learning [2][3], leading to insufficient realism in the rendering results. Here, we innovatively propose the use of a 3D GS renderer. Compared to traditional differentiable renderers, the gap between the 3D GS rendered image and the real image is significantly reduced, greatly alleviating gradient instability.
>
> 2. For the effectiveness of the second stage, visual ablation evidence can be found in the ablation study demo **demo_sCDj.mp4** (36s-1min11s) in the supplementary materials, which validates the effectiveness of our design. Our generated mesh is better aligned with the target video, and the mouth movements are free from exaggeration, unrealism, and noticeable noise. You can also refer to **Fig9** in the appendix, where I present the ablation results concerning the second stage. After the second stage, the generated mesh is better aligned with the GT image.
>
> 3. Why use the 3DGS renderer instead of a differentiable renderer: Although the 3DGS renderer may introduce minor artifacts in certain cases, it provides finer details that aid model learning. Compared to differentiable renderers [4], the 3DGS renderer offers a significant overall advantage in visual fidelity, greatly narrowing the gap between the rendered image and the real image. The comparison of 3DGS and differentiable renderers are detailed in Figure 11 in **supplementary material D** (also shown in the **demo_sCDj.mp4** (1min13s-1min23s)).
>
>
> ***
> > **Q3:**  The relationship with INFP is not fully clarified. MANGO also ultimately renders to 2D; does the main difference only lie in its use of 3DMM as an intermediate representation, rather than INFP's motion features? Thus, these two tasks are highly similar, and a visual comparison with INFP is crucial to demonstrate its advantage in motion controllability. Both methods employ dual-audio modules to link speech and motion features. How does MANGO's audio2motion model differ from and surpass INFP's?
>
> **A3:** Thank you for your question, which is greatly helpful for highlighting the unique characteristics of our paper.
>
> 1. MANGO focuses on Dual-Audio to Mesh generation for two-person digital humans, with the core objective of achieving high-fidelity dialogue-driven 3D mesh generation, unlike INFP which performs end-to-end single-person 2D video generation. Compared to 2D, 3D digital humans allow for explicit control over camera viewpoint, head movement, and expression parameters, which is the motivation for our choice of task. Therefore, the FLAME parameters here are not an intermediate representation but our core output, which is a fundamental difference from INFP.
>
> 2. INFP is not open-source, which prevents us from making a comparison, and it also cannot output explicit 3D mesh sequences; it models implicit motion features.
>
> 3. The difference from INFP: INFP uses two learnable memory banks to learn implicit motion features from speech, whereas we directly learn explicit, decoupled, and physically explainable motion parameters from speech. Furthermore, our structure is a clear 'encoding-feature fusion-decoding' unidirectional chain, essentially a regression task, while INFP's structure is more complex, involving the construction and learning of memory banks.
>
> 4. Compared to INFP's implicit motion space, MANGO's advantage at the representation level lies in its decoupling: the expression and pose parameters we obtain are decoupled and compatible with independent face shape and camera parameters. This decoupling makes our model easy to post-process and edit, while INFP's output is difficult to adjust once generated (any fine-tuning requires re-inference from the audio end). Additionally, since we provide motion labels, we can directly supervise the parameters, ensuring the generated motion parameters are more precise.
>
> - [2] SMIRK: 3D Facial Expressions through Analysis-by-Neural-Synthesis
> - [3] TEASER: Token Enhanced Spatial Modeling for Expression Reconstruction
> - [4] 3D Face Reconstruction with the Geometric Guidance of Facial Part Segmentation

---

> > ### Author Response · Authors · 2025-11-22
> > **Response to sCDj**
> >
> > ***
> > > **Q4:** The two claimed contributions, "conversational talking avatar generation and 2D image loss," seem weakly linked and more like two independent ideas. Furthermore, the statement "In conversational settings, lip movements become more complex due to interactive dynamics" (Line 80) lacks empirical evidence; single-speaker data may also exhibit similar complexity.
> >
> > **A4:** Thank you very much for your question and suggestion.
> > 1. We establish the link from the following perspective: We start with dual-audio as model input, and use the audio interaction module to generate the 3D mesh sequence. However, due to the lack of genuine 3D labels in the 3D domain, we must adopt the SOTA method from 3D reconstruction as our pseudo-label. We then observed the inaccuracy of this pseudo-label (Figure 2), which introduces a bias. To effectively mitigate this initial supervision bias, we propose introducing a strong constraint from the 2D ground-truth image during the model training phase. This ensures that the generated mesh aligns more closely with the corresponding 2D real image.
> >
> > 2. Thanks for your reminder and I apologize for the confusion. By stating that "In conversational settings, lip movements become more complex due to interactive dynamics," I intended to express that the audio-driven interactive motion generation task itself is inherently more complex than single-person listening or talking tasks due to the switching of speaking/listening states and the presence of interaction [1][5], thus making the modeling of lip movements more challenging for the model. I will revise our phrasing subsequently.
> >
> > ***
> > > **Q5:** The nomenclature Stage 1/2 and Training Phase 1/2 is confusing. For example, does Training Phase 1 (Stage 2 training) refer only to training the MG-Renderer on pGT meshes using the 2D image loss?
> >
> > **A5:** Thank you very much for your question; this is enormously helpful for enhancing the clarity of our paper. Stage primarily refers to the model components:
> > 1. Stage 1 is audio-to-mesh, and Stage 2 is mesh-to-image.
> > 2. Phase primarily refers to the training process: In Phase 1, we train Stage 1 and Stage 2 of the model separately. In Phase 2, we jointly and alternately train Stage 1 and Stage 2, allowing the image-level loss derived from Stage 2 to provide supervision for Stage 1.
> >
> > ***
> > > **Q6:** In Equation (3), $Z_{self}$ and $Z_{other}$ pass through the Transformer together, which seems inconsistent with the diagram in Figure 4.
> >
> > **A6:**
> > Thank you very much for your suggestion and reminder. As shown in Equation (3), the input to our Transformer is the concatenation of the audio features $H_{\text{self}}$ and $H_{\text{others}}$ of the interacting parties along the feature dimension, which is then fed into the Transformer. The purpose is to capture the long-range dependencies and complex interactive patterns between the two speakers. Regarding the data flow of $H_{\text{self}}$ in Figure 4, it actually has two flows (vertically displayed): one (upwards) is concatenated with $H_{\text{other}}$ along the feature dimension and then flows into the Transformer encoder; the other (horizontally to the right) is added to the interactive features output by the Transformer encoder.
> >
> >
> > ***
> > > **Q7:** Based on our understanding, the listener's expression in a dialogue should depend not only on the interlocutor's verbal content but also on the speaker's facial expressions. Is the speaker's state (or the speaker's visual features) considered as an additional input when modeling listening behavior?
> >
> > **A7:**
> > Thank you for your suggestion. Although adding the speaker's facial expressions could provide a supervision signal, this is not the core focus of the current task. As the next paradigm following audio-driven talking head tasks, our dual-audio driven talking-head places more emphasis on the model's interactive parsing of the audio, which is the core area that both single-person and dual-person talking-head research aims to improve. Furthermore, our micro-expression experiments mentioned previously also demonstrate that audio control is sufficient to model the speaker's facial expressions effectively.
> >
> > - [1] DualTalk: Dual-Speaker Interaction for 3D Talking Head Conversations
> > - [5] INFP: Audio-Driven Interactive Head Generation in Dyadic Conversations

---

### Official Review · Reviewer_6xaw · 2025-10-30

**Soundness:** 2
**Presentation:** 3
**Contribution:** 2
**Rating:** 2
**Confidence:** 5

**Summary:**

The paper proposes MANGO, a two-stage framework for generating natural, bidirectional 3D talking heads in multi-speaker conversational settings. Unlike prior work that focuses on either speaking or listening, MANGO aims to model fluid transitions between these states using dual-audio inputs and 2D photometric supervision to refine 3D motion. The authors also introduce MANGO-Dialog, a new dataset of 50+ hours of aligned 2D–3D conversational videos across 500+ identities. The core idea is to bypass the inaccuracies of pseudo-3D labels (from 3D face trackers) by using 2D image-level losses to guide 3D motion learning through a 3D Gaussian renderer.

**Strengths:**

1. Dual-audio fusion module enables speaker–listener disentanglement
The paper introduces a Dual-audio Interaction Module (DIM) that explicitly models conversational dynamics by fusing self and other speaker audio through a Transformer, followed by a residual connection with the self-audio. This design helps preserve speaker-specific lip-sync fidelity while allowing listener behaviors (e.g., subtle smiles, head nods) to be conditioned on the interlocutor’s speech. As shown in Fig. 7(a) and Table 2, removing this module leads to cross-contamination—e.g., the listener exhibits speaking-like mouth movements. This is a non-trivial contribution, as prior multi-speaker methods (e.g., DualTalk) do not explicitly model such asymmetric audio roles.

2. MANGO-Dialog: A large-scale, temporally aligned 2D–3D conversational dataset
The authors release MANGO-Dialog, comprising 50+ hours of dual-speaker videos across 500+ identities, with synchronized audio, pseudo-3D FLAME parameters (via Spectre), and refined camera poses. Crucially, clips are 30–120 seconds long, ensuring natural speaking–listening transitions—a rarity in existing datasets (e.g., VoxCeleb, HDTF are mostly single-speaker). The dataset also includes speaker diarization labels (via TackNet), enabling training of the speaking indicator. While the 3D labels are pseudo-ground truth (see Cons), the 2D–3D alignment and scale make this a valuable resource for future research in conversational avatars.

**Weaknesses:**

1. Inadequate 2D SOTA comparison

The paper compares its 2D output against SadTalker (2023), AniTalker (2024), and ARTalk (2025)—all of which are 3D-parameter-driven 2D renderers, not end-to-end 2D diffusion or neural rendering pipelines. It omits recent high-fidelity 2D talking head methods that achieve near-photorealism and strong lip-sync, such as:

VASA-1 (Microsoft, 2024): generates real-time, high-resolution, emotionally expressive talking faces from audio + single image.

OmniHuman-1: supports full-body, multi-view, and expressive control.

IF-MDM (2024): uses masked diffusion for coherent long-term 2D animation.

GaussianTalker / FlashAvatar: pure 3D Gaussian-based pipelines that may share architectural similarities with MANGO’s renderer but are not discussed or compared.

Without these comparisons, the claim of “superior 2D realism” is not convincingly supported—especially since MANGO’s 2D results (Fig. 6) show limited texture fidelity (e.g., blurry teeth, flat skin shading) compared to VASA-style outputs.

2. Missing comparison with industry-grade 3D pipelines

The 3D evaluation (Table 1) only includes academic methods (FaceFormer, CodeTalker, DualTalk, etc.). It excludes NVIDIA Audio2Face, which is:

*Widely used in production,

*Trained on high-quality 3D scans,

*Capable of real-time inference,

*Supports expression and viseme controls.

Given that MANGO claims “exceptional accuracy,” a comparison with Audio2Face on the same test set (even via qualitative side-by-side) would be essential to validate industrial relevance.

3. No explicit modeling of head pose dynamics or eye blinking

While the FLAME model includes head pose, the paper does not evaluate or visualize head motion quality. In Fig. 5–6, heads appear mostly static, suggesting the model may underutilize head pose variation—a key aspect of natural listening (e.g., nodding, tilting). Similarly, eye blinking is absent: FLAME does not model eyelids, and the renderer does not synthesize blinking. This leads to unnaturally fixed gazes, reducing perceived realism—especially in listening mode, where blink rate and gaze shifts are critical social signals. Previous methods such as DiffPoseTalk, Media2Face, already include the head-pose prediction and some also deliver natural eye blinking.

4. Limited expression control and variation

The method uses FLAME’s expression parameters (ψ ), but the paper provides no analysis of non-mouth expressions (e.g., brow raises, smiles, frowns). While Fig. 6 shows some smiling, it’s unclear whether this is audio-driven or coincidental. There is no user control over expression intensity or type, and no disentanglement between speech-driven and emotion-driven motion. This limits applications requiring emotional or stylistic control.

5. 3D labels derived from noisy 2D-to-3D lifting

The dataset’s 3D labels come from Spectre, which the paper itself critiques (Fig. 2) for over-smoothing or exaggerating lip motion. This creates a fundamental supervision bottleneck: even with 2D-lifted refinement, the initial motion prior is biased. The authors claim their output sometimes exceeds the pseudo-GT mesh (Fig. 5, 9), but this is not quantified (e.g., via human preference or 2D re-projection error vs. GT image). Without ground-truth 3D scans (e.g., from multi-view capture), the true 3D accuracy remains unverifiable.

6. Ablation studies are missing from the demo video

The paper includes strong ablation results (Table 2, Fig. 7), but the supplementary demo video (presumably linked in submission) does not visualize these variants (e.g., w/o DIM, w/o two-stage). This makes it hard for reviewers/users to perceptually validate the claimed improvements. For a method relying on subtle conversational cues, visual ablation is essential.

**Questions:**

1. How is Fig. 2 generated?

Fig. 2 shows “over-smoothed” (orange) and “exaggerated/noisy” (blue) 3D lip motion curves compared to a “real” red curve, with visual insets of misaligned meshes. However, the paper does not specify:

What is the ground-truth reference for the red curve? Is it manually annotated lip keypoints, or derived from high-fidelity 3D scans?

Which 3D reconstruction methods produced the orange and blue curves? Are they Spectre, DECA, 3DDFA-v3, etc.?

Are these curves from real conversational data (like MANGO-Dialog) or from single-speaker datasets?

Without this, the figure risks being illustrative rather than empirical, weakening the motivation for 2D-lifted supervision.

2. The paper claims MANGO sometimes outperforms pseudo-GT meshes (e.g., Fig. 5, 9). But how is this quantified?

The visual examples in Fig. 5 and Fig. 9 suggest that MANGO’s mesh aligns better with the 2D ground-truth image than the pseudo-GT mesh from Spectre. However:

Is there a 2D re-projection error (e.g., L1 distance between rendered mesh and GT image) comparing MANGO vs. Spectre?

Have you conducted a user study where humans judge which mesh (Spectre vs. MANGO) better matches the GT video?

If MANGO is “better than pseudo-GT,” does that imply the pseudo-GT is a poor training target—and if so, why not use 2D-only supervision from the start?

This is central to the paper’s core claim but remains anecdotal.

3. The speaking indicator I_self is assumed to be perfectly known. How does performance degrade under realistic diarization errors?
The method uses a binary speaking indicator derived from TackNet (Sec 3.4), which is likely near-perfect on curated clips. But in real-world deployment:

What happens if I_self flips state 10% or 20% of the time (common in overlapping speech)?

Is the model robust to missing or delayed indicators?

Could the model infer the speaking state from audio alone, removing reliance on external diarization?

This affects practical applicability, yet no ablation on indicator noise is provided.

4. The ablation in Table 2 shows “Ours (+two stage)” has higher LVE/MVE than the “+jaw pose” variant. Why does adding 2D supervision increase 3D vertex error?

In Table 2:

The “+jaw pose” row: LVE = 0.235,

The full “Ours (+two stage)” row: LVE = 0.122,

But in the MANGO-Dialog column of Table 1, the full model reports LVE = 1.741, which is much higher than the ablation’s 0.122. This suggests a unit or normalization inconsistency.

Are the ablation metrics computed on mouth vertices only (as in LVE definition), while Table 1 uses full mesh?

Or is there a scaling difference (e.g., mm vs. normalized units)?

Please clarify the metric definitions and scales across tables to ensure comparability.

---

> ### Author Response · Authors · 2025-11-22
> **Response to 6xaw**
>
> Dear Reviewer 6xaw,
>
> Thank you for your comprehensive suggestions on our paper. We hope our clarifications address your concerns.
> ***
> > **Q1:** Insufficient comparison with 2D SOTA techniques. Specifically, comparison is missing with end-to-end 2D diffusion methods VASA-1 (Microsoft, 2024), OmniHuman-1, IF-MDM, and 3DGS methods GaussianTalker / FlashAvatar.
>
> **A1:** Thanks for your suggestions.
>
> 1.	Regarding the end-to-end 2D diffusion methods (VASA-1, OmniHuman-1, IF-MDM):
> Our work focuses on dialogue-driven 3D mesh generation, aiming to obtain explicit and controllable 3D expression and pose parameters. This fundamentally differs from the task performed by end-to-end 2D methods like VASA-1, OmniHuman-1, and IF-MDM, whose output is direct pixel data and does not provide controllable 3D geometry. Furthermore, none of these three recent works have made their code public, which currently prevents direct quantitative comparison.
>
> 2. Regarding the 3DGS-based methods (GaussianTalker / FlashAvatar):
>    * GaussianTalker is a single-person, single-model approach. The model is trained based on the speaking portrait video of a specific individual and cannot be generalized to an arbitrary person. Our model, however, is ID-agnostic and can generalize to unseen identities.
>     * FlashAvatar focuses on the efficient reconstruction of an animatable digital avatar from a single monocular video. Its task is not audio-driven digital human generation, resulting in a significant disparity from our core objective.
>
> 3. Regarding the term “Superior Realism” mentioned in the our paper's Abstract: We believe there may have been a misunderstanding of our intent. The term "superior realism" was specifically used in the abstract to refer to the quality of the generated 3D mesh, not the final rendered image fidelity. We will revise our description in the manuscript to avoid misinterpretation by the readers.
>
> ***
> > **Q2:** Missing comparison with industry-grade 3D pipelines NVIDIA Audio2Face
>
> **A2:** Thank you again for your thorough consideration. We did attempt to conduct a comparison with NVIDIA Audio2Face, but faced the following difficulties.
> 1. NVIDIA Audio2Face officially provides only three fixed IDs, which prevents us from conducting a fair comparison using the identities in our dialogue test dataset.
> 2. NVIDIA Audio2Face remains a single-person, audio-driven model, making it challenging to migrate its application to multi-speaker tasks.
> 3. The official model only outputs CSV files. Since the official open-source content does not disclose the rendering pipeline, it is extremely difficult for us to achieve qualitative result visualization within a short timeframe.

---

> ### Author Response · Authors · 2025-11-22
> **Response to 6xaw**
>
> > **Q3:** There is no explicit modeling of head pose dynamics or blinking motions. The quality of head movement is not evaluated; heads in Figures 5-6 appear mostly static, blinking is missing, and the realism of the listening state is reduced.
>
> **A3:** Thank you very much for your detailed suggestions regarding the performance of our model.
> 1. To quantitatively evaluate the quality of our model's head movements and the diversity of facial expressions, we calculated the SI Diversity (SID) metric [1]. This metric quantifies diversity by applying k-means clustering to the motion sequences in the feature space and calculating the entropy of the cluster assignment histogram. A higher SID value indicates richer and more diverse motion patterns in the generated animation. The results in the table below demonstrate that our model achieves the best diversity across pose, expression (exp), and jaw movements.
>
>     | Method | SID-pose $\uparrow$ | SID-exp $\uparrow$ | SID-jaw $\uparrow$ |
>     | :--- | :--- | :--- | :--- |
>     | DiffPoseTalk | 1.47 | 1.96 | 1.73 |
>     | DualTalk | 1.48 | 2.17 | 1.98 |
>     | **Ours** | **1.53** | **2.23** | **2.26** |
>
> 2. We understand your observation that the heads in Figures 5-6 appear mostly static. This is primarily because head motion is difficult to convey in a single static image. Furthermore, during a conversation, overly dramatic head movements from a listener, especially when the speaker's tone is relatively calm, can actually reduce realism. We have provided more examples of generated head and blinking movements in **demo_6xaw.mp4** (0s-56s) in the supplementary materials for your review.
>
> 3. The blinking motion itself is already encompassed within the FLAME expression parameter space, and the videos we generate successfully synthesize natural blinking movements, which you can observe in our generated videos in **demo_6xaw.mp4** (0s-56s). Furthermore, due to the ambiguous relationship among speech, blinking motions, and gaze direction, we did not explicitly model their interdependencies. In addition, to further demonstrate the realism of our generated movements (especially in the listening state), we measured the naturalness of the listener's movements using both the quantitative Fréchet Distance (FD) metric and a user study. Both evaluations demonstrate that the movements generated by our method possess higher realism. Specifically, we found that:"
>     * We utilize the Frechet Distance (FD) [1] to calculate the distributional distance between the generated motions and the ground-truth motions in the feature space; a lower distance indicates stronger motion realism. For both the speaker and listener segments, our method demonstrates superior performance in expression, jaw, and pose compared to the baseline methods. These experiments validate the effectiveness of our diffusion-based 3D motion generation and 2D-level supervision.
>
>         | Method | S-FD $\downarrow$ (EXP) | S-FD $\downarrow$ (JAW $\times 10^3$) | S-FD $\downarrow$ (POSE $\times 10^2$) | L-FD $\downarrow$ (EXP) | L-FD $\downarrow$ (JAW $\times 10^3$) | L-FD $\downarrow$ (POSE $\times 10^2$) |
>         | :--- | :---: | :---: | :---: | :---: | :---: | :---: |
>         | DiffPoseTalk | 23.86 | 2.89 | 3.61 | 18.39 | 3.56 | 4.79 |
>         | DualTalk | **21.91** | 3.06 | 3.83 | 12.94 | 2.12 | 3.23 |
>         | **Mango(Ours)** | 22.37 | **2.75** | **3.54** | **11.93** | **1.99** | **2.78** |
>
>         *Note: L stands for Listener segments, and S stands for Speaker segments. The following is the same*
>
>     * We invited 15 participants to use the Mean Opinion Score (MOS) protocol to rate the realism [1] of the test set's listening and speaking segments across four dimensions: pose naturalness, expression richness, visual quality, and audio-lip synchronization. The results indicate that our method achieved higher average scores across virtually all dimensions compared to the baseline methods.
>         | Methods | L-Visual Quality $\uparrow$ | L-Expression Richness $\uparrow$ | L-Pose Naturalness $\uparrow$ | S-Lip Sync Accuracy $\uparrow$| S-Visual Quality $\uparrow$| S-Expression Richness $\uparrow$| S-Pose Naturalness $\uparrow$|
>         | :--- | :---: | :---: | :---: | :---: | :---: | :---: | :---: |
>         | CodeTalker | 2.6 | 2.6 | 2.8 | 2.1 | 1.8 | 2.2 | 1.3 |
>         | DiffPoseTalk | 2.3 | 3.4 | 3.5 | **4.4** | 3.8 | 3.8 | 3.9 |
>         | DualTalk | 3.5 | 3.8 | 3.2 | 4.0 | 3.5 | 3.6 | 3.7 |
>         | Mango(Ours) | **3.9** | **3.9** | **4.0** | 4.3 | **4.1** | **3.9** | **4.0** |
>
> - [1] DualTalk: Dual-Speaker Interaction for 3D Talking Head Conversations

---

> ### Author Response · Authors · 2025-11-22
> **Response to 6xaw**
>
> > **Q4:** Limited expressiveness and control. The paper lacks analysis of non-mouth expressions, does not provide user control over expression intensity or type, nor does it decouple speech-driven motion from emotion-driven motion.
>
> **A4:** Thank you very much for your suggestions, which provide highly meaningful directions for our future work.
> 1. Our current task primarily focuses on two-person dialogue, and while our model implicitly models a small subset of the relationships between speech and expression (in **demo_6xaw.mp4** (3min51s-4min41s) in the supplementary materials), subtle expressions such as smiling or frowning were not the main focus of our research.
> 2. Modeling micro-expressions and emotion is a challenging direction, requiring more fine-grained modeling and more diverse datasets [2]. Several existing works specifically focus on single-person emotional digital humans [2][3], and the difficulties become even more pronounced in interactive digital humans. If the goal is to specifically model expressions vividly or achieve emotion disentanglement, a much more comprehensive and enriched dataset would be necessary.
>
> ***
> > **Q5:** The 3D labels derived from Spectre create a fundamental supervision bottleneck. Superiority over pseudo-Ground Truth (GT) is claimed but not quantified (e.g., reprojection error or human preference), and without 3D scan data, true 3D supervision remains unverifiable.
>
> **A5:** Thank you for your question, I will further refine my statement.
> 1. Why did we choose Spectre? 3D face tracking remains an active research area, and the current state of 3D reconstruction is generally limited due to the lack of large-scale, genuine 3D scan data. However, among existing methods, Spectre utilizes multi-frame lip-reading losses for temporal optimization, enabling its reconstructed 3D speaking head to approximate the audio-lip synchronization of the original video as closely as possible, compared to most methods that focus on single-frame RGB reconstruction. Therefore, Spectre remains a mainstream 3D reconstruction method in the current SOTA audio-driven 3D face generation [1]. Considering all factors, we chose Spectre.
>
> 2. Despite Spectre's advantages, its pseudo-GT meshes do exhibit certain biases, such as: mouth opening being sometimes exaggerated (e.g., Figure 2, Row 2 in Figure 5, and Figure 9 in the paper), insufficient mouth closure in the listening state (Figure 2, Row 2, column 8), and potential mesh penetration during lip pursing. To effectively mitigate this initial supervision bias, we propose introducing a strong constraint from the 2D ground-truth image during the model training phase. This ensures that the generated mesh, when projected into the 2D image space, aligns with the real image pixels to the maximum extent possible.
>
> 3. We have quantified the claim that 'our output is sometimes superior to the pseudo-GT' using the 2D reprojection error in our response (A8) to Q8.
>
> 4. In the supplementary **demo_6xaw.mp4** (57s-2min13s), we provide a comparison between several results reconstructed by Spectre and our generated results. This comparison shows that the Spectre reconstructions often exhibit overly exaggerated expressions, whereas our results appear to be more aligned with the real video across most frames.
>
> ***
> > **Q6:** Lack of intuitive video demonstration for the ablation of individual modules.
>
> **A6:** Thank you very much for your suggestion. We have demonstrated the effect of each ablated module by showing the generation results before and after ablation in **demo_6xaw.mp4** (2min13s-3min51s) in the supplementary materials.
>
> 1. Regarding the ablation of the DIM (Dual-person Interaction Modeling) module: It is evident from the demo video that without the DIM module, the model struggles to accurately distinguish between the speaking and listening states. This difficulty leads to the model failing to suppress mouth movements when the speaker is silent, resulting in unnecessary 'random' jaw movements. Simultaneously, some emotional cues in the speech (e.g., smiles) are also not captured.
>
> 2. Regarding the ablation of the Indicator: Similarly, the inclusion of the Indicator leads to a more natural transition between the speaking and listening states, resulting in a clear overall visual improvement.
>
> 3. Regarding the ablation of the Two-Stage Optimization: The lack of the second stage of optimization results in excessively large mouth opening amplitudes and some visually noticeable, unnatural expression sudden changes (abrupt changes). Conversely, with the inclusion of the second stage, the exaggerated mouth opening is mitigated, and the overall dialogue state appears more realistic.
>
> - [1] DualTalk: Dual-Speaker Interaction for 3D Talking Head Conversations
> - [2] EmoTalk: Speech-Driven Emotional Disentanglement for 3D Face Animation
> - [3] LaughTalk: Expressive 3D Talking Head Generation With Laughter

---

> ### Author Response · Authors · 2025-11-22
> **Response to 6xaw**
>
> > **Q7:** What does the red curve in Figure 2, representing the ground-truth reference? Which 3D reconstruction methods do the generated orange and blue curves represent? where does these curves originate from?
>
> **A7:** Thank you very much for your questions.
> 1. The red curve is derived by calculating the distance between the corresponding key points of the manually annotated upper and lower lips, which effectively describes the actual mouth movement.
>
> 2. The orange curve represents the average distance between the corresponding key points of the upper and lower lips, calculated after projecting the 3D mesh reconstructed by Teaser back into 2D space. Methods like this often suffer from inaccurate jaw pose, leading to an over-smooth appearance. The blue curve shows the results from Spectre. As evident from this curve, Spectre exhibits rich mouth movements, but they are often overly exaggerated and contain noise.
>
> 3. This curve sequence was randomly selected by us from real dialogue data.
>
> 4. For detailed information regarding the curves, please refer to the answer for the next question. (A8)
>
> ***
>
> > **Q8:** How is MANGO's superiority over the pseudo-GT mesh quantified? Was a human preference study performed to judge which mesh (Spectre vs. MANGO) better matches the GT video? why not use 2D-only supervision from the start?
>
> **A8:**  Thank you very much for your questions.
>
> 1. We apologize for previously omitting the detailed quantification process. We provide the supplement below, the calculation procedure steps are as follows:
>     * Get projected 2D keypoints: We project the 3D vertices reconstructed by Mango and Spectre, respectively, onto the 2D space via camera projection to obtain their corresponding 2D planar coordinates, $X_m$ and $X_s$.
>     * Ground-Truth Keypoints: We manually annotate the keypoints on the corresponding GT images to obtain $X_{gt}$
>     * Lip Distance Metric Calculation: We statistically calculate the average distance between the corresponding keypoints of the upper and lower lips for each frame, serving as a metric to measure the degree of mouth opening and closing. The calculation process is as follows:
>         $$D_{\text{Lip}, t} = \frac{1}{N} \sum_{i=1}^{N} \sqrt{(x^U_{i, t} - x^L_{i, t})^2 + (y^U_{i, t} - y^L_{i, t})^2}$$
>         where $x^U_{i, t}$, $x^L_{i, t}$, $y^U_{i, t}$, and $y^L_{i, t}$ represent the x-coordinate and y-coordinate of the $i$-th pair of upper and lower lip keypoints, respectively, at time $t$.
>     * Curve Generation: We calculate this metric for every frame across the three sets of keypoints (Mango, Spectre, and GT) to obtain $D_{\text{Lip}, t}^{mango}$, $D_{\text{Lip}, t}^{spectre}$, and $D^{gt}_{\text{Lip}, t}$, which are then used to plot the curves shown in Figure 2.
>     * Mean Absolute Error (MAE) Calculation: Statistically, we calculate the Mean Absolute Error (MAE) across the test sequences using the following formula. Finally, we obtained the following result:
>         | Evaluation Metric | Spectre-Reconstruction | Mango-Generation|
>         | :---: | :---: | :---: |
>         | **$\text{MAE}_{\text{Lip}}$** $\downarrow$ | 9.68 | **6.23** |
>
>
> 2. To evaluate visual fidelity, we recruited 20 participants for a user study. Participants were asked to select the sample that was visually more consistent with the 2D image ground-truth from 10 pairs of clips (our generated results vs. Spectre's reconstruction). The test results showed that our generated results achieved a user preference of nearly 80%.
>
>     | Evaluation Metric | Spectre-Reconstruction | Mango-Generation|
>     | :---: | :---: | :---: |
>     | **User Preference** | Approx. 20% | **Approx. 80%** |
>
> 3. Our core objective is to generate accurate 3D Mesh from dialogue audio. Therefore, all our training strategies are designed around this goal. In the initial stage, the direct introduction of 3D geometric supervision provides the model with a clear and correct optimization direction, which greatly facilitates the model's efficient learning of decoupled 3D motion parameters from the speech input. In contrast, if we rely purely on 2D supervision, it would, on the one hand, hinder the establishment of the mapping from speech to 3D Mesh geometry. On the other hand, some motion details might be overfitted in the implicit rendering stage [4], ultimately impeding the model from learning a precise and robust geometric motion representation.
>
> 4. In the supplementary **demo_6xaw.mp4** (57s-2min13s), we provide a comparison between several results reconstructed by Spectre and our generated results. This comparison shows that the Spectre reconstructions often exhibit overly exaggerated expressions, whereas our results appear to be more aligned with the real video across most frames.
>
> - [4] SMIRK: 3D Facial Expressions through Analysis-by-Neural-Synthesis

---

> > ### Author Response · Authors · 2025-11-22
> > **Response to 6xaw**
> >
> > > **Q9:**  How does the performance degrade under realistic speaker diarization errors? What happens if $I_{self}$ flips state for 10% or 20% of the time (which is common during speech overlap)? Is the model robust to missing or delayed indicators?
> >
> > **A9:**
> > Thank you for your reminder. We adopted consecutive misattribution noise to simulate misattribution errors in real-world scenarios. Specifically, given a segment of speech with length $L$, we perform an XOR operation ($\oplus$) between the original speaker mask $A$ and a consecutive noise mask $\Delta A'$ to obtain the noisy mask $\tilde{A}'$:
> >     $$\tilde{A}' = A \oplus \Delta A'$$
> >     where $\Delta A'$ is generated by randomly selecting a consecutive segment of length $L_{\text{flip}}$ in $A$ and flipping its 0/1 values, where:$$L_{\text{flip}} = \lceil \alpha \cdot L \rceil, \alpha \in [0.1, 1]$$
> > 1. Experimental results show that the model's performance did not degrade significantly even with an error rate where the segment length factor $\alpha$ reached $0.3$ (i.e., $30\%$ of the consecutive segments were flipped), which demonstrates the strong robustness of our model. This anti-interference capability primarily stems from the model's reliance on longer-term speech features to infer the dialogue's logic and context, which effectively mitigates the negative impact caused by short-term, consecutive $0/1$ misattribution errors.
> > 2. Additionally, we observe that the indicator prediction accuracy of the speech separation model we utilize exceeds $70\%$ (specifically, $90.8\%$ [5]), which suggests our model is robust in most scenarios. We present the curve illustrating the change in model performance as the consecutive flip segment length factor $\alpha$ varies in the Appendix C.
> >
> > ***
> > > **Q10:**  The ablation study in Table 2 shows that the LVE/MVE of the 'Ours (+two stage)' variant is higher than the '+jaw pose' variant. Is this metric calculated solely on the mouth vertices, or is there a scaling difference involved?
> >
> > **A10:** Thank you very much for your meticulous review. We sincerely apologize, as the MVE and LVE metrics for the 'Ours (+two stage)' row in Table 2 were mistakenly swapped. The correct values should be MVE: 1.225 and LVE: 0.174. (The LVE calculation method is consistent between Table 1 and Table 2—it is calculated solely on the mouth vertices, resulting in the same numerical value, but we used different magnitudes). We will correct the paper and unify the magnitudes subsequently.
> >
> > - [5] Is Someone Speaking? Exploring Long-term Temporal Features for Audio-visual Active Speaker Detection

---

### Official Review · Reviewer_sYZJ · 2025-11-02

**Soundness:** 3
**Presentation:** 3
**Contribution:** 3
**Rating:** 6
**Confidence:** 4

**Summary:**

The paper proposes MANGO, a two-stage framework for natural, multi-speaker 3D talking head generation. Stage-1 predicts FLAME parameters (facial expression and articulated head/jaw pose) directly from dual-speaker audio, explicitly modeling speaking/listening interaction. Stage-2 renders the predicted 3D motion with a 3D-Gaussian splatting–based image generator and uses 2D photometric/perceptual losses to lift supervision back to the 3D motion, alternating training between the motion and the renderer. On a new multi-speaker conversational dataset, the method reports improved 3D motion accuracy and 2D visual/lip-sync scores versus recent baselines.

**Strengths:**

- Modeling both speakers’ audio and the role switch (speaking vs. listening) is well motivated and aligns with emerging conversation-aware talking-head research. This is a non-trivial step beyond speaker-only driving.

- Alternating 2D-lifted supervision is elegant and plausible. Using a fast differentiable renderer (3D Gaussians) to refine motion predicted in Stage-1 is technically sound and explains the observed improvement in mouth articulation/expressiveness.

- Evaluation with community-recognized metrics. Reporting LSE-C/LSE-D alongside image-space metrics aligns with established practice in audio-visual lip-sync evaluation.

**Weaknesses:**

- Limited analysis of listening behaviors. The qualitative figures suggest better non-verbal feedback (nodding, smiles), but there is little targeted measurement of listener' realism beyond global metrics including more diverse non-verbal signals. Consider role conditioned metrics or human studies that separately score speaking vs. listening segments.

- Ablations could isolate Stage-2’s contribution more sharply. It would help to report identical Stage-1 models trained (a) without any 2D-lifted refinement, (b) with only photometric vs. only perceptual losses, and (c) with/without Gaussian renderer fine-tuning.

**Questions:**

1. How sensitive is Stage-1 to errors in active-speaker detection and speech overlaps? Any quantitative robustness test (e.g., synthetic noise or mis-attribution)?

2. Can the model generalize to unseen speakers and to diverse emotions (e.g., laughter, surprise)? A small cross-emotion test would be informative. Please refer relevant work [1].

[1] LaughTalk: Expressive 3D Talking Head Generation with Laughter, https://arxiv.org/pdf/2311.00994

---

> ### Author Response · Authors · 2025-11-22
> **Response to sYZJ**
>
> Dear Reviewer sYZJ,
>
> We are grateful for your positive review and valuable comments. We hope our response fully resolves your concerns.
> ***
> > **Q1:** Analysis of listener behavior is limited. Quantitative measurement of listener realism is lacking. Please consider using role-conditioned metrics or a user study to score speaking and listening segments separately.
>
> **A1:** Thanks for your suggestions.  We evaluated the generated segments for both the speaker and the listener separately by calculating the motion realism metric FD and conducting a user study. Our model successfully demonstrates good preservation of listener authenticity across both evaluation methods. Specifically:
>
> 1. We utilize the Fréchet Distance (FD) [1] to calculate the distribution distance between the generated motions and real motions in the feature space, where a lower distance indicates higher motion realism. For both the speaking and listening segments, our method generally outperforms the compared baselines in terms of expression, jaw, and pose.
>
>     | Method | S-FD $\downarrow$ (EXP) | S-FD $\downarrow$ (JAW $\times 10^3$) | S-FD $\downarrow$ (POSE $\times 10^2$) | L-FD $\downarrow$ (EXP) | L-FD $\downarrow$ (JAW $\times 10^3$) | L-FD $\downarrow$ (POSE $\times 10^2$) |
>     | :--- | :---: | :---: | :---: | :---: | :---: | :---: |
>     | DiffPoseTalk | 23.86 | 2.89 | 3.61 | 18.39 | 3.56 | 4.79 |
>     | DualTalk | **21.91** | 3.06 | 3.83 | 12.94 | 2.12 | 3.23 |
>     | **Mango(Ours)** | 22.37 | **2.75** | **3.54** | **11.93** | **1.99** | **2.78** |
>
> *Note: L stands for Listener segments, and S stands for Speaker segments. The following is the same*
>
> 2.	We invited 15 participants and used the Mean Opinion Score (MOS) rating protocol. They were asked to separately score the authenticity of the listening segments and the speaking segments in the test set across four dimensions: pose naturalness, expression richness, visual quality, and audio-lip synchronization [1]. The results indicate that our method achieved higher average scores across most dimensions.
>
>     | Methods | L-Visual Quality $\uparrow$ | L-Expression Richness $\uparrow$ | L-Pose Naturalness $\uparrow$ | S-Lip Sync Accuracy $\uparrow$| S-Visual Quality $\uparrow$| S-Expression Richness $\uparrow$| S-Pose Naturalness $\uparrow$|
>     | :--- | :---: | :---: | :---: | :---: | :---: | :---: | :---: |
>     | CodeTalker | 2.6 | 2.6 | 2.8 | 2.1 | 1.8 | 2.2 | 1.3 |
>     | DiffPoseTalk | 2.3 | 3.4 | 3.5 | **4.4** | 3.8 | 3.8 | 3.9 |
>     | DualTalk | 3.5 | 3.8 | 3.2 | 4.0 | 3.5 | 3.6 | 3.7 |
>     | Mango(Ours) | **3.9** | **3.9** | **4.0** | 4.3 | **4.1** | **3.9** | **4.0** |
>
>
> ***
> > **Q2:**  Provide clearer ablations to validate the contribution of Stage-2. It would be helpful to report results trained with the same Stage-1 model, specifically including:
> (a) Without any 2D-lifted refinement. (b) Using photometric loss only and perceptual loss only. (c) With/without Gaussian renderer fine-tuning.
>
> **A2:** Thank you for your valuable suggestions. We have added more detailed ablation studies concerning Stage-2. The results confirm that utilizing both photometric loss and perceptual loss while performing alternating fine-tuning of the Gaussian renderer (corresponding to our final model) significantly enhances the effectiveness of the 2D supervision, thereby demonstrating advantages across all metrics.
>
>
> | Ablation Study | MVE $\downarrow$ | LVE $\downarrow$ | MTM $\downarrow$ |
> | :--- | :---: | :---: | :---: |
> | **mango** | **1.225** | **1.741** | **4.015** |
> | w/o any 2D-lifted refinement | 1.507 | 2.351 | 4.147 |
> | w/o perceptual loss | 1.392 | 2.048 | 4.124 |
> | w/o photometric loss | 1.406 | 2.126 | 4.092 |
> | w/o Gaussian renderer fine-tuning | 1.473 | 2.309 | 4.036 |
>
> - [1] DualTalk: Dual-Speaker Interaction for 3D Talking Head Conversations

---

> ### Author Response · Authors · 2025-11-22
> **Response to sYZJ**
>
> > **Q3:**  How sensitive is Stage-1 to errors in active-speaker detection and speech overlaps? Any quantitative robustness test (e.g., synthetic noise or mis-attribution)?
>
> **A3:**
> Thank you for your reminder. We adopted consecutive misattribution noise to simulate misattribution errors in real-world scenarios. Specifically, given a segment of speech with length $L$, we perform an XOR operation ($\oplus$) between the original speaker mask $A$ and a consecutive noise mask $\Delta A'$ to obtain the noisy mask $\tilde{A}'$:
> $$\tilde{A}' = A \oplus \Delta A'$$
> where $\Delta A'$ is generated by randomly selecting a consecutive segment of length $L_{\text{flip}}$ in $A$ and flipping its 0/1 values, where:$$L_{\text{flip}} = \lceil \alpha \cdot L \rceil, \alpha \in [0.1, 1]$$
> * Experimental results show that the model's performance did not degrade significantly even with an error rate where the segment length factor $\alpha$ reached $0.3$ (i.e., $30\%$ of the consecutive segments were flipped), which demonstrates the strong robustness of our model. This anti-interference capability primarily stems from the model's reliance on longer-term speech features to infer the dialogue's logic and context, which effectively mitigates the negative impact caused by short-term, consecutive $0/1$ misattribution errors.
> * Additionally, we observe that the indicator prediction accuracy of the speech separation model we utilize exceeds $70\%$ (specifically, $90.8\%$ [2]), which suggests our model is robust in most scenarios. We present the curve illustrating the change in model performance as the consecutive flip segment length factor $\alpha$ varies in the Appendix C.
>
> ***
> > **Q4:**  Can the model generalize to unseen speakers and to diverse emotions (e.g., laughter, surprise)? A small cross-emotion test would be informative.
>
> **A4:**
> Thank you for your question. We will provide a clearer explanation.
> 1. Our model generalizes to unseen identities (or unseen subjects), as all individuals in our test dataset are identities not present in the training set.
> 2. Modeling emotion is a relatively difficult direction in 3D digital human research, requiring finer-grained modeling and more diverse datasets [3]. Several papers specifically focus on research into single-person affective digital humans [4, 5], and the difficulty becomes even more pronounced for interactive digital humans. Although our current work does not specifically address or analyze emotion, we have observed that our method implicitly models certain emotional expressions to some extent (as seen in **demo_sYZJ.mp4** in the supplementary materials). For instance, the agent exhibits a smile while speaking during the 23-25 seconds of the demo.mp4 in the Supplementary Material. In the future, we hope to commit to mining datasets rich in emotional content to study interactive emotional digital humans.
>
> - [2] Is Someone Speaking? Exploring Long-term Temporal Features for Audio-visual Active Speaker Detection
> - [3] EmoTalk: Speech-Driven Emotional Disentanglement for 3D Face Animation
> - [4] LaughTalk: Expressive 3D Talking Head Generation With Laughter
> - [5] FlowVQTalker: High-Quality Emotional Talking Face Generation through Normalizing Flow and Quantization

---

### Author Response · Authors · 2025-11-25
**We remain available for any further clarification**

Dear Reviewers,

We sincerely thank you for the valuable time and thoughtful feedback invested in the evaluation of our submission. Your comments are indicative of a deep engagement with our work and have provided us with clear directions for significantly enhancing the paper's quality.

We have provided detailed, point-by-point clarifications and modification descriptions in our rebuttal letter in response to all the concerns you raised. We genuinely hope that these responses, along with the changes made in the revised manuscript, will effectively address your questions and facilitate a better understanding of our core contributions.

As the discussion period continues, we wish to reaffirm our readiness to provide further supplementary material. If you feel any aspect requires additional support for your final assessment—such as deeper theoretical analysis, further explanation of key experimental results, supplementary ablation studies, or any other supporting information—please do not hesitate to let us know. We would be very happy and able to promptly provide any extra details and data you may need.

Thank you again for your insightful input and for the indispensable assistance you have provided in improving this research work.

---

### Author Response · Authors · 2025-11-27
**Common Response to All Reviewers**

We thank all reviewers for their valuable feedback. We are greatly encouraged by the acknowledgment that our paper is "is well motivated"(`sYZJ`), "a non-trivial step"(`sYZJ/6xaw`), "novel framework"(`X7SN`), "elegant and plausible"(`sYZJ`),"technically sound"(`sYZJ`), "a valuable contribuution"(`6xaw`, `X7SN`), "a novel and ambitious direction"(`sCDj`), "comprehensive evaluation"(`X7SN`).

In our responses, we have carefully followed the reviewers' comments and provided additional clarifications, experimental results, and discussions. We believe these additions make the paper more complete and better support the effectiveness of our proposed methods.  Below is a summary of my response.

***
**Task and method**
* We provided a clearer definition of the task itself, distinguishing it from end-to-end diffusion methods and 3D Gaussian-based methods for single-person talking head generation (`6xaw`-Q1Q2, `sCDj`-Q3, `X7SN`-Q4).
* We explained the difference between our task and methods specifically designed for emotion and micro-expression control (`sYZJ`-Q4, `6xaw`-Q4).
* We further explained the necessity of the two-stage approach for this task and discussed why the Gaussian renderer was chosen over traditional differentiable renderers in the second stage (`sCDj`-Q2Q4).
* We supplemented some technical details and explanations (`6xaw`-Q7Q8, `X7SN`-Q5).

**Regarding the experiment**
* We additionally conducted a comprehensive quantitative and qualitative analysis of listening behaviors (`sYZJ`-Q1Q2, `6xaw`-Q3).
* We performed a comprehensive evaluation and provided a visual demonstration of the model's expressiveness, covering its capability to model head nodding, smiling micro-expressions, and emotional conveyance in speech (`sYZJ`-Q4, `6xaw`-Q4, `sCDj`-Q1).
* We supplemented the experiments by providing additional video-level demonstrations for the ablation study of each module (`6xaw`-Q6).
* We further included supplementary ablation experiments to validate the effectiveness of the proposed methodology (`sYZJ`-Q2).
* We conducted additional robustness testing on the indicator (`sYZJ`-Q3, `6xaw`-Q9).

**About future work and discussions**
* We performed an analysis of the scalability of our model (`X7SN`-Q2).
* We incorporated the emotional modeling of interactive 3D digital humans into our future work (`sYZJ`-Q4, `6xaw`-Q4).

***
Furthermore, we have prepared **some video demos** in the supplementary material for the reviewers' convenience (please refer to my response for the location of the corresponding content).
We also remain committed to addressing any further questions or concerns from the reviewers promptly.

Best regards,

The Authors

---

### Meta-Review · Area_Chair_Dv8M · 2025-12-20

**Summary:**

The decision was driven by concerns regarding the paper's scope and its conceptual novelty, leading to an unfavorable middle ground.

1) Limited Scope and Utility. The method suffers from a critical trade-off deficit, being constrained in both domains:

- Constrained for 3D (Reviewer 6xaw): Lacks essential features for a natural avatar (e.g., explicit blinking/gaze) and fails to benchmark against industry-grade tools (Audio2Face).
- Low-Fidelity for 2D (Reviewer X7SN): The output quality is significantly lower than SOTA pixel-generators (VASA-1, HALLO3), limiting its use where photorealism is required.

2. Weak Conceptual Novelty. The core technical solution is perceived as incremental rather than a breakthrough:

- Task Composition (Reviewer X7SN): The framework is seen as a simple combination of existing modules (diffusion and dual-audio processing), lacking deep architectural novelty.

- 2D-Lifted Enhancement is a Necessary Patch (Reviewer sCDj): The primary innovation is viewed as a specialized fix to correct the acknowledged noise in the pseudo-3D training data (Spectre), rather than a general conceptual advance in the field.

Despite efforts in the rebuttal to address specific concerns, the persistent problems of limited scope and weak conceptual novelty remained insufficiently resolved, rendering the paper unable to demonstrate the adequate new ideas, community contribution, or strong empirical results necessary for acceptance

**Reviewer Concerns:**

The authors effectively addressed nearly all specific technical and evaluation concerns by providing new quantitative data (e.g., FD metrics, LME, ablation tables) and clarifying method. However, core issues related to scope and visual competitiveness remain partially unresolved.

1. Addressed & Resolved Concerns. The authors successfully provided evidence and clarification for the following:

- Evaluation Metrics: Added new quantitative tables and MOS studies specifically for listening behavior and provided the correct scale for the LVE table inconsistency. (sYZJ, sCDj, 6xaw)

- Method Robustness: Confirmed the superiority of MANGO's mesh over pseudo-GT using LME and demonstrated robustness to Diarization Errors up to 20%. (6xaw, sYZJ)

- Clarity: Fully clarified the distinction between "Stages" and "Phases" and explicitly defined the "2D-Lifted Enhancement" as a training strategy. (X7SN, sCDj)

- Ablations: Provided a sharp ablation table for the Stage 2 renderer. (sYZJ)

2. Outstanding Concerns. The most critical outstanding issues revolve around the method's positioning and competitive results:

- Missing 2D SOTA Comparison: Authors conceptually defended the missing comparisonby arguing MANGO's scope is explicit 3D parameter output for control/utility, while SOTA 2D methods output pixels. The conceptual defense is accepted, but the visual fidelity gap remains. (6xaw, X7SN)

- Weak Conceptual Novelty (Partially Mitigated): Reviewer skepticism persists that the work is a "simple combination" (Task Composition), even though the authors argued novelty lies in the interactive listener model and the specialized 2D-lifted strategy. (X7SN, sCDj)

- Relatively Lower Visual Quality (Outstanding): The low visual rating persists and is acknowledged by the authors as an accepted trade-off for achieving explicit 3D geometry control and faster performance (100+ FPS). (X7SN)

**Reviewer Scores:**

While the positive reviewer is expected to maintain or potentially raise their score, the negative reviews will likely persist because the core deficiencies regarding visual quality and the 2D SOTA comparison gap were not fully mitigated. Consequently, the paper may struggle to gain the necessary strong advocate for final acceptance.

---

### Decision · Program_Chairs · 2026-01-26

Reject